# How accurate are estimates of glacier ice thickness? Results from ITMIX, the Ice Thickness Models Intercomparison eXperiment

Daniel Farinotti[1,2], Douglas J. Brinkerhoff[3], Garry K.C. Clarke[4], Johannes J. Fürst[5], Holger Frey[6], Prateek Gantayat[7], Fabien Gillet-Chaulet[8], Claire Girard[9], Matthias Huss[1,10], Paul W. Leclercq[11], Andreas Linsbauer[6,10], Horst Machguth[6,10], Carlos Martin[12], Fabien Maussion[13], Mathieu Morlighem[9], Cyrille Mosbeux[8], Ankur Pandit[14], Andrea Portmann[2], Antoine Rabatel[8], RAAJ Ramsankaran[14], Thomas J. Reerink[15], Olivier Sanchez[8], Peter A. Stentoft[16], Sangita Singh Kumari[14], Ward J.J. van Pelt[17], Brian Anderson[18], Toby Benham[19], Daniel Binder[20], Julian A. Dowdeswell[19], Andrea Fischer[21], Kay Helfricht[21], Stanislav Kutuzov[22], Ivan Lavrentiev[22], Robert McNabb[3,11], G. Hilmar Gudmundsson[12], Huilin Li[23], and Liss M. Andreassen[24]

[1]Laboratory of Hydraulics, Hydrology and Glaciology (VAW), ETH Zurich, Zurich, Switzerland
[2]Swiss Federal Institute for Forest, Snow and Landscape Research (WSL), Birmensdorf, Switzerland
[3]Geophysical Institute, University of Alaska Fairbanks, Fairbanks, AK, USA
[4]Department of Earth, Ocean and Atmospheric Sciences, University of British Columbia, Vancouver BC, Canada
[5]Institute of Geography, Friedrich-Alexander-University Erlangen-Nuremberg (FAU), Erlangen, Germany
[6]Department of Geography, University of Zurich, Zurich, Switzerland
[7]Divecha Centre for Climate Change, Indian Institute of Science, Bangalore, India
[8]Institut des Géosciences de l'Environnement (IGE), Université Grenoble Alpes, CNRS, IRD, Grenoble, France
[9]Department of Earth System Science, University of California Irvine, Irvine, CA, USA
[10]Department of Geosciences, University of Fribourg, Fribourg, Switzerland
[11]Department of Geosciences, University of Oslo, Oslo, Norway
[12]British Antarctic Survey, Natural Environment Research Council, Cambridge, UK
[13]Institute of Atmospheric and Cryospheric Sciences, University of Innsbruck, Innsbruck, Austria
[14]Department of Civil Engineering, Indian Institute of Technology, Bombay, India
[15]Institute for Marine and Atmospheric Research (IMAU), Utrecht University, Utrecht, The Netherlands
[16]Arctic Technology Centre ARTEK, Technical University of Denmark, Kongens Lyngby, Denmark
[17]Department of Earth Sciences, Uppsala University, Uppsala, Sweeden
[18]Antarctic Research Centre, Victoria University of Wellington, Wellington, New Zealand
[19]Scott Polar Research Institute, University of Cambridge, Cambridge, UK
[20]Central Institute for Meteorology and Geodynamics (ZAMG), Vienna, Austria
[21]Institute for Interdisciplinary Mountain Research, Austrian Academy of Sciences, Innsbruck, Austria
[22]Laboratory of Glaciology, Institute of Geography, Russian Academy of Science, Moscow, Russia
[23]State Key Laboratory of Cryospheric Sciences, Tian Shan Glaciological Station, CAREERI, CAS, Lanzhou, China
[24]Norwegian Water Resources and Energy Directorate (NVE), Oslo, Norway

*Correspondence to:* Daniel Farinotti (daniel.farinotti@ethz.ch)

**Abstract.** Knowledge of the ice thickness distribution of glaciers and ice caps is an important prerequisite for many glaciological and hydrological investigations. A wealth of approaches has recently been presented for inferring ice thickness from characteristics of the surface. With the Ice Thickness Models Intercomparison eXperiment (ITMIX) we performed the first coordinated assessment quantifying individual model performance. A set of 17 different models showed that individual ice thickness estimates can differ considerably – locally by a spread comparable to the observed thickness. Averaging the results of multiple models, however, significantly improved the results: On average over the 21 considered test cases, comparison against direct ice thickness measurements revealed deviations in the order of $10 \pm 24\%$ of the mean ice thickness ($1\sigma$ estimate). Models relying on multiple data sets – such as surface ice velocity fields, surface mass balance, or rates of ice thickness change – showed high sensitivity to input data quality. Together with the requirement of being able to handle large regions in an automated fashion, the capacity of better accounting for uncertainties in the input data will be a key for an improved next generation of ice thickness estimation approaches.

## 1 Introduction

The ice thickness distribution of a glacier, ice cap, or ice sheet is a fundamental parameter for many glaciological applications. It determines the total volume of the ice body, which is crucial to quantify water availability or sea-level change, and provides the link between surface and subglacial topography, which is a prerequisite for ice-flow modelling studies. Despite this importance, knowledge about the ice thickness of glaciers and ice caps around the globe is limited – a fact linked mainly to the difficulties in measuring ice thickness directly. To overcome this problem, a number of methods have been developed to infer the total volume and/or the ice thickness distribution of ice masses from characteristics of the surface.

Amongst the simplest methods, so-called *scaling approaches* are the most popular (see Bahr et al., 2015, for a recent review). These approaches explore relationships between the area and the volume of a glacier (e.g. Chen and Ohmura, 1990; Bahr et al., 1997), partially including other characteristics such as glacier length or surface slope (e.g. Lüthi, 2009; Radić and Hock, 2011; Grinsted, 2013). Such approaches, however, yield estimates of the mean ice thickness and total volume of a glacier only.

Methods that yield distributed information about the ice thickness generally rely on theoretical considerations. Nye (1952), for example, noted that for the case of an idealized glacier of infinite width, ice thickness can be calculated from the surface slope using estimates of basal shear stress and assuming perfect plastic behaviour. Nye (1965) successively extended the considerations to valley glaciers of idealized shapes, whilst Li et al. (2012) additionally accounted for the effect of side drag from the glacier margins. Common to these three approaches is the assumption of a constant and known basal shear stress. Haeberli and Hoelzle (1995) were the first suggesting that the latter can be estimated from the glacier elevation range, and the corresponding parametrization has been used in a series of recent studies (e.g. Paul and Linsbauer, 2011; Linsbauer et al., 2012; Frey et al., 2014).

Early approaches that take into account mass conservation and ice flow dynamics go back to Budd and Allison (1975) and Rasmussen (1988), whose ideas were further developed by Fastook et al. (1995) and Farinotti et al. (2009). The latter approach was successively extended by Huss and Farinotti (2012), who presented the first globally complete estimate for the

ice thickness distribution of individual glaciers. Alternative methods based on more rigorous inverse-modelling, on the other hand, have often focused on additionally inferring basal slipperiness together with bedrock topography (e.g. Gudmundsson et al., 2001; Thorsteinsson et al., 2003; Raymond-Pralong and Gudmundsson, 2011; Mosbeux et al., 2016).

In the recent past, the number of methods aiming at estimating the ice thickness distribution from characteristics of the surface has increased at a rapid pace. Methods have been presented that include additional data such as surface velocities and mass balance (e.g. Morlighem et al., 2011; McNabb et al., 2012; Clarke et al., 2013; Farinotti et al., 2013; Huss and Farinotti, 2014; Gantayat et al., 2014; Brinkerhoff et al., 2016), as well as approaches that make iterative use of more complex forward models of ice flow (e.g. van Pelt et al., 2013; Michel et al., 2013, 2014), or non-physical methods based on neural network approaches (Clarke et al., 2009; Haq et al., 2014). This development has led to a situation in which a wealth of approaches is potentially available, but no assessment comparing the relative strengths and weaknesses of the models exists.

Against this background, the Working Group on glacier ice thickness estimation, hosted by the International Association of Cryospheric Sciences (`www.cryosphericsciences.org`), launched the *Ice Thickness Models Intercomparison eXperiment (ITMIX)*. The experiment aimed at conducting a coordinated comparison between models capable of estimating the ice thickness distribution of glaciers and ice caps from surface characteristics. Emphasis was put on evaluating the model performance when no a-priori information on actual ice thickness is provided. This was to focus on the most widespread application of such models; that is, the estimation of the ice thickness of an unmeasured glacier.

This article presents both the experimental setup of ITMIX and the results of the intercomparison. The accuracy of individual approaches is assessed in a unified manner, and the strengths and shortcomings of individual models are highlighted. By doing so, ITMIX not only provides quantitative constraints on the accuracies that can be expected from individual models, but also aims at setting the basis for developing a new generation of improved ice thickness estimation approaches.

## 2 Experimental setup

ITMIX was conducted as an open experiment, with a call for participation posted on the email distribution list "Cryolist" (`http://cryolist.org/`) on 13 October 2015. Individual researchers known to have developed a method for estimating glacier ice thickness were invited personally. Upon registration, participants were granted access to the input data necessary for the experiment and the corresponding set of instructions.

The input data referred to the surface characteristics of a predefined set of 21 test cases (see next section, Tab. 1, and Fig. 1) and participants were asked to use these data for generating an estimate of the corresponding ice thickness distribution. Results were collected, and compared to direct ice thickness measurements.

No prior information about ice thickness was provided, and the participants were asked not to make use of published ice thickness measurements referring to the considered test cases for model calibration. This was to mimic the general case in which the ice thickness distribution for unmeasured glaciers has to be estimated. The compliance to the above rule relied on honesty.

Participants were asked to treat as many test cases as possible, and to consider data-availability (cf. next section and Tab. 1) as the only factor limiting the number of addressed cases. Details on the considered test cases and the participating models are given in Sections 3 and 4 respectively. An overview of the solutions submitted to the experiment is given in Table 2.

## 3   Considered test cases and data

The considered test cases included 15 glaciers and 3 ice caps for which direct ice thickness measurements are available, and 3 synthetically generated glaciers virtually "grown" over known bedrock topographies (more detailed information below). The real-world test cases (see Fig. 1 for geographical distribution) were chosen to reflect different glacier morphologies (cf. Tab. 1) and different climatic regions, whilst the synthetic test cases were included to have a set of experiments for which all necessary information is perfectly known. Since most published approaches for estimating ice thickness were developed for applications

on mountain glaciers and smaller ice caps, ice sheets where not included in the experiment.

For each test case, the input data provided to the ITMIX participants included at a minimum (a) an outline of the glacier or ice cap, and (b) a gridded digital elevation model (DEM) of the ice surface. Further information was provided on a case-by-case basis depending on data availability, including the spatial distribution of the (i) surface mass balance (SMB), (ii) rate of ice thickness change ($\partial h/\partial t$), and (iii) surface flow velocity. An overview of the data available for individual test cases and the

corresponding data sources is given in Table 2 and Table 1, respectively.

For the real-world test cases, and whenever possible, temporal consistency was ensured between individual data sets. Glacier outlines and DEMs were snapshots for a given point in time, whereas SMB, $\partial h/\partial t$, and velocity fields generally referred to multi-year averages for an epoch as close as possible to the corresponding DEM. Glacier-wide estimates of surface velocities were not available for any of the considered cases. For obtaining a possibly complete coverage, velocities from separate sources

were therefore merged, which often led to discontinuities along the tile margins.

Ice thickness measurements were only used for quantifying model performance but were not distributed to the ITMIX participants. Bedrock elevations were obtained by subtracting observed ice thicknesses from surface elevations, and the bedrock was assumed to remain unchanged over time. The time periods the individual data sets are referring to are given in Supplementary Table S1. Note that no specific information about the uncertainties associated to individual measurements were available.

Reported uncertainties for ice thickness measurements, however, are typically below 5% (Plewes and Hubbard, 2001).

The synthetic test cases were generated by "growing" ice masses over known bedrock topographies with the *Elmer/Ice* ice flow model (Gagliardini et al., 2013). To do so, selected deglacierized areas located in the European Alps were extracted from local high-resolution DEMs (product *DHM25* by the Swiss Federal Office of Topography), and the flow model forced with a prescribed SMB field. The SMB field was either generated by prescribing an equilibrium-line altitude and two separate

SMB elevation gradients for the accumulation and ablation zone (test cases "Synthetic1" and "Synthetic2"), or by constructing the field through a multiple linear regression between SMB and terrain elevation, slope, aspect, curvature, and local position (test case "Synthetic3"). In the latter case, the individual regression parameters were defined arbitrarily but such to ensure a plausible range for the resulting SMB field. The Elmer/Ice simulations were stopped after the formation of a glacier judged to

**Table 1.** Overview of the test cases considered in ITMIX. Glacier type follows the *GLIMS classification guidance* (Rau et al., 2005). "Calv" indicates whether the glacier or ice cap is affected by calving (x) or not (-). Further abbreviations: A = glacier area; SB = simple basin; CB = compound basin; mnt. = mountain; OL = glacier outline, DEM = digital elevation model of the glacier surface, SMB = surface mass balance, Vel. = surface ice flow velocity, $\partial h/\partial t$ = rate of ice thickness change, H = ice thickness measurements, Unpub. = Unpublished data by. References to the data are given.

| Test case | Type | Calv | A (km$^2$) | Available data and source |
|---|---|---|---|---|
| Academy | Ice cap | x | 5587.2 | OL, DEM, H: Dowdeswell et al. (2002) |
| Aqqutikitsoq | SB valley gl. | - | 2.8 | OL, DEM, H: Marcer et al. (in review) |
| Austfonna | Ice cap | x | 7804.8 | OL, DEM: Moholdt and Kääb (2012); $\partial h/\partial t$, SMB: Unpub. G. Moholdt; Vel.: Dowdeswell et al. (2008); H: Dowdeswell et al. (1986) |
| Brewster | SB mountain gl. | - | 2.5 | OL: LINZ (2013); DEM: Columbus et al. (2011); SMB: Anderson et al. (2010); Vel.: Unpub. B. Anderson; H: Willis et al. (2009) |
| Columbia | CB valley gl. | x | 937.1 | OL, DEM, H: McNabb et al. (2012) |
| Devon | Ice cap | x | 14015.0 | OL, DEM, H: Dowdeswell et al. (2004); Vel.: Unpub. GAMMA[1] |
| Elbrus | Crater mnt. gl. | - | 120.8 | OL, $\partial h/\partial t$, H: Unpub. RAS[2]; DEM: Zolotarev and Khrkovets (2000); SMB: WGMS (1991-2012) |
| Freya | SB valley gl. | - | 5.3 | OL, DEM, H: Unpub. ZAMG[3]; SMB: Hynek et al. (2015) |
| Hellstugubreen | CB valley gl. | - | 2.8 | OL: Andreassen et al. (2008); DEM, SMB, $\partial h/\partial t$: Andreassen et al. (2016); Vel.: Unpub. NVE[4]; H: Andreassen et al. (2015) |
| Kesselwandferner | SB mountain gl. | - | 4.1 | OL, DEM: Fischer et al. (2015); SMB: Fischer et al. (2014); H: Fischer and Kuhn (2013) |
| Mocho | Crater mnt. gl. | - | 15.2 | OL, H: Geostudios LTA (2014); DEM: ASTER GDEM v2[5]; SMB: Unpub. M. Schaefer |
| North Glacier | SB valley gl. | - | 7.0 | OL, DEM, H: Wilson et al. (2013); Vel.: Unpub. G. Flowers |
| South Glacier | SB valley gl. | - | 5.3 | OL, DEM, H: Wilson et al. (2013); SMB: Wheler et al. (2014); Vel.: Flowers et al. (2011) |
| Starbuck | CB outlet gl. | x | 259.7 | OL, H: Farinotti et al. (2014); DEM: Cook et al. (2012) |
| Tasman | CB valley gl. | - | 100.3 | OL: LINZ (2013); DEM: Columbus et al. (2011); SMB, Vel.: Unpub. B. Anderson; H: Anderton (1975) |
| Unteraar | CB valley gl. | - | 22.7 | OL, DEM, $\partial h/\partial t$, SMB: Unpub. VAW-ETHZ[6]; Vel.: Vogel et al. (2012); H: Bauder et al. (2003) |
| Urumqi | SB mountain gl. | - | 1.6 | OL, DEM, SMB, H: Wang et al. (2016) |
| Washmawapta | Cirque mnt. gl. | - | 0.9 | OL, DEM, H: Sanders et al. (2010) |
| Synthetic1 | CB valley gl. | - | 10.3 | OL, DEM, SMB, Vel., $\partial h/\partial t$, H: Unpub. C. Martin and D. Farinotti |
| Synthetic2 | CB mountain gl. | - | 35.3 | OL, DEM, SMB, Vel., $\partial h/\partial t$, H: Unpub. C. Martin and D. Farinotti |
| Synthetic3 | Ice cap | - | 89.9 | OL, DEM, SMB, Vel., $\partial h/\partial t$, H: Unpub. C. Martin and D. Farinotti |

[1] GAMMA Remote Sensing Research and Consulting AG, Gümligen, Switzerland; contact person T. Strozzi

[2] Russian Academy of Sciences, Institute of Geography, Moscow, Russia; contact person S. Kutuzov

[3] Zentralanstalt für Meteorologie und Geodynamik (ZAMG), Vienna, Austria; contact person D. Binder

[4] Norwegian Water Resources and Energy Directorate (NVE), Oslo, Norway; contact person L.M. Andreassen

[5] ASTER GDEM is a product of NASA and METI

[6] Laboratory of Hydraulics, Hydrology and Glaciology (VAW), ETH Zurich, Zurich, Switzerland; contact person A. Bauder

**Table 2.** Overview of provided and used data, as well as test cases considered by individual models. Names of ice caps are flagged with an asterisk (*). Models are named after the modeller submitting the results; alternative model identifiers that have been used in the literature are given in parenthesis. The category refers to the classification provided in Section 4 and includes (1) minimization approaches, (2) mass conserving approaches, (3) shear-stress based approaches, (4) velocity-based approaches, and (5) other approaches. Abbreviations: OL+DEM= Glacier outline and digital elevation model of the surface; SMB = surface mass balance; Vel. = surface ice flow velocity; $\partial h/\partial t$ = rate of ice thickness change. For "Vel.", a distributed field of flow speeds (s) and flow directions (d), or individual point measurements (p) were provided. "x" (".") indicates that the given information was (not) provided/used. In the columns "Data used", "(x)" indicates that the information was used when available, but that it is not strictly necessary for model application. References for the data source are given in Table 1.

| Category | Model / Provided data | Academy* | Aqqutikitsoq | Austfonna* | Brewster | Columbia | Devon* | Elbrus | Freya | Hellstugubreen | Kesselwandferner | Mocho | North Glacier | South Glacier | Starbuck | Tasman | Unteraar | Urumqi | Washmawapta | Synthetic1 | Synthetic2 | Synthetic3 | TOTAL cases | OL+DEM | SMB | Vel. | $\partial h/\partial t$ |
|---|---|---|---|---|---|---|---|---|---|---|---|---|---|---|---|---|---|---|---|---|---|---|---|---|---|---|---|
| | OL+DEM | x | x | x | x | x | x | x | x | x | x | x | x | x | x | x | x | x | x | x | x | x | | | | | |
| | SMB | . | . | x | x | . | . | x | x | x | x | x | . | x | . | x | x | x | . | x | x | x | | | | | |
| | Vel. | . | . | sd | p | . | sd | . | . | p | . | . | p | p | . | s | sd | . | . | sd | sd | sd | | | | | |
| | $\partial h/\partial t$ | . | . | x | . | . | . | x | . | x | . | . | . | . | . | . | x | . | . | x | x | x | | | | | |
| 1 | Brinkerhoff-v2 | . | . | . | x | . | . | x | x | x | . | . | x | . | . | x | x | . | . | x | x | x | 10 | x | x | (x) | . |
| 1 | Fuerst | . | . | x | . | . | . | . | . | . | . | . | . | . | . | . | x | . | . | x | x | x | 5 | x | x | x | x |
| 1 | VanPeltLeclercq | . | . | . | x | . | . | x | x | x | x | x | . | . | . | . | x | . | . | x | x | x | 10 | x | x | (x) | . |
| 2 | Farinotti (ITEM) | x | x | x | x | x | x | x | x | x | x | x | x | x | x | x | x | x | x | x | x | x | 21 | x | . | . | . |
| 2 | GCbedstress | x | x | . | x | . | . | x | x | x | x | x | . | . | . | x | x | x | x | x | x | x | 15 | x | (x) | . | (x) |
| 2 | Huss (HF-model) | x | x | x | x | x | x | x | x | x | x | x | x | x | x | x | x | x | x | x | x | x | 21 | x | . | . | . |
| 2 | Maussion (OGGM) | x | x | x | x | x | x | x | x | x | x | x | x | x | . | x | x | x | x | x | x | . | 19 | x | . | . | . |
| 2 | Morlighem | . | . | . | . | . | . | x | x | x | . | . | x | x | . | x | x | . | . | x | x | x | 10 | x | x | (x) | . |
| 3 | Linsbauer (GlabTop) | x | x | x | x | x | x | x | x | x | x | x | x | x | x | x | x | x | x | x | x | x | 21 | x | . | . | . |
| 3 | Machguth (GlabTop2) | x | x | . | x | x | x | x | x | x | x | x | x | x | x | x | x | x | x | x | x | x | 20 | x | . | . | . |
| 3 | RAAJglabtop2 | . | . | . | . | . | . | . | . | . | . | . | x | . | x | x | . | . | . | . | x | . | 4 | x | . | . | . |
| 4 | Gantayat | . | . | x | . | . | x | . | . | . | . | . | . | . | . | x | x | . | . | x | x | x | 7 | x | . | x | . |
| 4 | Gantayat-v2 | . | . | x | . | . | x | . | . | . | . | . | . | . | . | x | x | . | . | x | x | x | 7 | x | . | x | . |
| 4 | RAAJgantayat | . | . | . | x | . | . | . | . | . | . | . | x | . | . | x | x | . | . | x | . | . | 5 | x | . | x | . |
| 4 | Rabatel | . | . | . | . | . | . | . | . | . | . | . | . | . | . | . | x | . | . | . | . | . | 1 | x | x | x | . |
| 5 | Brinkerhoff | . | . | . | . | . | . | . | . | . | . | . | . | . | . | . | . | . | . | x | x | x | 3 | x | x | x | x |
| 5 | GCneuralnet | . | x | . | . | . | . | . | . | x | x | x | x | . | x | x | . | . | . | x | x | x | 10 | x | . | . | . |
| **TOTAL models** | | 6 | 7 | 7 | 9 | 5 | 7 | 6 | 9 | 9 | 10 | 8 | 9 | 9 | 4 | 11 | 15 | 8 | 6 | 16 | 15 | 13 | **189** | | | | |

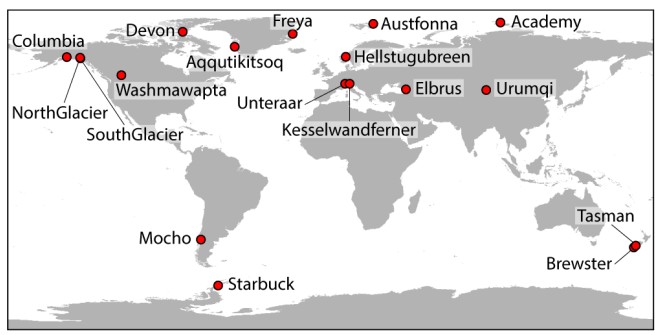

**Figure 1.** Overview of the considered real-world test cases. Note that some names are shortened for convenience (Academy = Academy of Sciences Ice Cap; Devon = Devon Ice Cap; Mocho = Glaciar Mocho-Choshuenco; Unteraar = Unteraargletscher; Urumqi = Urumqi Glacier No. 1).

be of suitable size and shape, and the corresponding $\partial h/\partial t$ and surface velocity fields were extracted. No sliding at the glacier
base was assumed, and all three resulting geometries were close to steady state. Note that, to avoid numerical instabilities, the DEM used for prescribing the bedrock topography had to be smoothed significantly. For anonymising the individual locations, the original coordinates were removed, and the individual tiles arbitrarily rotated, and shifted in elevation.

All data provided as input to the ITMIX participants, as well as the results submitted by individual models, will be provided as an electronic supplement to this article. The direct ice thickness measurements were additionally included in the Glacier
Thickness Database (GlaThiDa) version 2 (WGMS, 2016).

## 4   Participating models

The ITMIX call for participation was answered by 13 research groups having access to 15 different models in total. Two modelling approaches were used twice, with two independent implementations stemming from two different groups; 9 models were published prior to the call; 1 model consisted of a modification of an existing approach; and 5 models were previously
unpublished. In total, thus, 17 different models submitted individual solutions (Tab. 2).

The 17 approaches providing individual solutions can be classified into five different categories: (1) Approaches casting ice thickness inversion as a minimization problem (*minimization approaches*), (2) approaches based on mass conservation (*mass conserving approaches*), (3) approaches based on a parametrization of basal shear stress (*shear-stress based approaches*), (4) approaches based on observed surface velocities (*velocity-based approaches*), and (5) other approaches. The principle of
each of the five categories is briefly described hereafter. A more detailed description, including information about parameter choices, is found in the Supplementary Material (Supplementary Section S1). The supplementary description is exhaustive for unpublished approaches and is held at a minimum for published ones.

## 4.1 Minimization approaches

Methods within this category formulate the problem of ice thickness inversion as a minimization problem. They do so by defining a cost function that penalizes the difference between a modelled and an observable quantity. Typically, the observable quantity includes the elevation of the glacier surface (e.g. Leclercq et al., 2012; Michel et al., 2013; van Pelt et al., 2013), which can be obtained from a surface DEM. Given an initial guess for the subglacial bedrock topography, a forward model for glacier ice flow is then used to predict the observable quantity. The difference between model and observation is subsequently used to update the model, and the procedure is repeated iteratively to minimize the cost function. The forward model can be of any type, generally considers mass conservation (see next section), and often relies on a higher-order representation of ice dynamics. Three models of this category participated in ITMIX:

"**Brinkerhoff-v2**" (Brinkerhoff, unpublished; see Supplementary Section S1.2 for details) includes three terms in the cost function. The first term quantifies the difference between modelled and observed surface elevations; the second penalizes strong spatial variations in bedrock elevations; and the third is used to impose zero ice thickness outside the glacier boundaries. If available, surface flow velocities are used to additionally invert for the basal traction field. The forward model is based on the Blatter-Pattyn approximation to the Stokes' equations (Pattyn, 2003).

"**VanPeltLeclercq**" (adapted from van Pelt et al., 2013, Supplementary Section S1.17) has a cost function based on the difference of modelled and observed surface elevation as well. In contrast to "Brinkerhoff-v2", which evaluates the cost function for steady-state surfaces, this approach allows for transient surface geometries to be taken into account. If available, the mismatch between calculated and observed surface velocities is used for both stopping the iteration procedure and for optimizing the model parameters affecting basal sliding and deformational flow. The implemented forward model "SIADYN" is part of the ICEDYN package (Sec. 3.3 in Reerink et al., 2010), relies on the vertically integrated shallow ice approximation (e.g. Hutter, 1983), and includes Weertman-type sliding (Huybrechts, 1991).

"**Fuerst**"(Fürst et al., unpublished; Supplementary Section S1.4) differs from the two above approaches in that the cost function is not linked to surface elevations. Instead, the function penalizes (i) negative thickness values, (ii) the mismatch between modelled and observed surface velocities, (iii) the mismatch between modelled and observed SMB, and (iv) strong spatial variations in ice thickness or surface velocities. The forward model is based on Elmer/Ice (Gagliardini et al., 2013) and the mass conservation approach of Morlighem et al. (2011).

## 4.2 Mass conserving approaches

Methods appertaining to this category are based on the principle of mass conservation. If ice is treated as an incompressible medium, the corresponding continuum equation states that the ice flux divergence $\nabla \cdot q$ has to be compensated by the rate of ice thickness change $\frac{\partial h}{\partial t}$ and the climatic mass balance $\dot{b}$:

$$\nabla \cdot q = \frac{\partial h}{\partial t} - \dot{b}. \tag{1}$$

The methods of this category estimate the distribution of both $\frac{\partial h}{\partial t}$ and $\dot{b}$, and use that estimate to quantify the glacier's mass turnover along the glacier. The mass flux is then converted into ice thickness by prescribing some constitutive relation. Most often, an integrated form of Glen's flow law (Glen, 1955) is used. The corresponding equation, solved for ice thickness, is then generally formulated as:

$$h = \sqrt[n+2]{\frac{q}{2A} \cdot \frac{n+2}{(f\rho g \sin\alpha)^n}}, \tag{2}$$

where $h$ is glacier ice thickness, $q$ the mean specific ice volume flux, $A$ the flow rate factor, $n$ Glen's flow law exponent, $\rho$ the ice density, $g$ the gravitational acceleration, $\alpha$ the surface slope, and $f$ a factor accounting for valley shape, basal sliding, and parameter uncertainty. To avoid infinite $h$ for $\alpha$ tending to zero, a minimal surface slope is often imposed, or $\alpha$ is averaged over a given distance. Based on theoretical considerations (Kamb and Echelmeyer, 1986), this distance should correspond to 10-20 times the ice thickness. In most cases, the ice thickness is first inferred along prescribed ice flow lines, and then distributed across the glacier or ice cap by choosing a suitable interpolation scheme. Five of the models participating in ITMIX belong to this category:

"**Farinotti**" (Farinotti et al., 2009, also referred to as *ITEM* in the literature; Supplementary Section S1.3) evaluates Eq. 2 for manually digitized "ice flow catchments" and along manually predefined ice flow lines. The ice volume flux across individual cross sections is estimated by integrating the SMB field of the corresponding upstream area. The method was the first suggesting that the necessity of a steady-state assumption can be circumvented when directly estimating the difference $\dot{b} - \partial h / \partial t$, rather than imposing constraints on the two terms separately. Many of the approaches within this and other categories have adopted this idea.

"**Maussion**" (Maussion et al., unpublished; Supplementary Section S1.12) is based on the same approach as Farinotti et al. (2009). By relying on the *Open Global Glacier Model version 0.1.1* (*OGGM v0.1.1*; Maussion et al., 2017), however, it fully automatises the method, thus making it applicable at larger scales. Automatisation is achieved by generating multiple flowlines according to the methods presented in Kienholz et al. (2014). The major difference between "Maussion"/OGGM and the approaches "Farinotti" or "Huss" is that SMB is not prescribed as a linear function of elevation but with a temperature-index model driven by gridded climate data (Marzeion et al., 2012).

"**Huss**" (Huss and Farinotti, 2012, *HF-model*; Supplementary Section S1.9) extends Eq. 2 to account for additional factors such as basal sliding, longitudinal variations in the valley shape factor, frontal ablation, and the influence of ice temperature and the climatic regime. The latter is achieved by imposing site-specific parameters. A major difference compared to other models in this category is that all calculations are performed on elevation bands. Mean elevation-band thickness is then extrapolated to a spatially distributed field by considering local surface slope and the distance from the glacier margin. The approach was the first ice thickness model that was applied to the global scale.

"**GCbedstress**" (Clarke et al., 2013, Supplementary Section S1.7) shares many conceptual features with Farinotti et al. (2009) as well, but differs in its implementation. Manually delineated flowsheds are transversely dissected by ladder-like "rungs" representing flux gates. Ice flow discharges – derived from integration of the mass contribution from the upstream area – are then applied to intervening cells by interpolation. "Raw" ice thickness is derived from Eq. 2 and the final ice thickness

is smoothed by minimizing a cost function that negotiates a tradeoff between accepting the raw estimates or maximizing the smoothness of the solution.

"**Morlighem**" (Morlighem et al., 2011, Supplementary Section S1.13) was originally designed to fill gaps between ground-penetrating radar measurements over ice sheets. As such, it was cast as an optimization problem minimizing the misfit between observed and modelled thicknesses. Since no such measurements were provided within ITMIX, the method was applied without the minimization scheme. The method is thus purely based on mass conservation. Ice thickness is computed by requiring the ice flux divergence to be balanced by the rate of thickness change and the net surface and basal mass balances (cf. Eq. 1). For the test cases for which no ice velocities were provided, the shallow ice approximation (see below) was used together with an assumption of no-sliding to convert the computed ice mass flux into ice thickness.

## 4.3 Shear-stress based approaches

Methods of this category rely on the shallow ice approximation (e.g Fowler and Larson, 1978). In the latter, the relation

$$h = \frac{\tau}{\rho g \sin \alpha} \tag{3}$$

is assumed to hold true everywhere, from which it follows that knowledge of the basal shear stress $\tau$ allows for the ice thickness to be determined. Most existing approaches estimate $\tau$ from the empirical relation proposed by Haeberli and Hoelzle (1995), which relates $\tau$ to the elevation range of a glacier. The denominator of the right-hand side of the equation often includes an additional factor $f$, with a similar meaning as described for Eq. 2. The models of this category mostly differ from the ones in the last section in that they do not account for mass conservation. Three such approaches participated in ITMIX:

"**Linsbauer**" (Linsbauer et al., 2009, 2012, *GlabTop*; Supplementary Section S1.10) was the first proposing to use the empirical relation by (Haeberli and Hoelzle, 1995) to solve Eq. 3. This is done by considering manually digitized branchlines, and determining $\alpha$ within 50 m elevation bins. An ice thickness distribution is then obtained by interpolating the so-obtained ice thickness along several branchlines.

"**Machguth**" (Frey et al., 2014, *GlabTop2*; Supplementary Section S1.11) is based on the same concept, but overcomes the need of manually drawing branchlines by applying the relation at randomly selected grid cells. During this process, $\alpha$ is determined from the average slope of all grid cells within a predefined elevation buffer. The final ice thickness distribution is derived from interpolation of the randomly selected points and the condition of zero ice thickness at the glacier margin. The procedure by which the random points are selected has an influence on the shape of the obtained bedrock topography.

"**RAAJglabtop2**" (re-implemented from Frey et al., 2014, Supplementary Section S1.15) is an independent re-implementation of the "Machguth" model. Individual differences in terms of coding solutions may exist but were not assessed during the experiment.

## 4.4 Velocity-based approaches

As for models in Section 4.2, models in this category are based on an integrated form of Glen's flow law (Glen, 1955). Differently as in Eq. 2, however, the flow law is either expressed as

$$h = \frac{n+1}{2A} \frac{u_{\mathrm{s}} - u_{\mathrm{b}}}{\tau^n}, \tag{4}$$

– where $u_{\mathrm{s}}$ and $u_{\mathrm{b}}$ are the surface and basal ice flow velocities, respectively – or such to replace $q$ in Eq. 2 with the depth-averaged profile velocity $\overline{u}$ (since $q = \overline{u} h$). An assumption relating $u_{\mathrm{s}}$ to $u_{\mathrm{b}}$ or $\overline{u}$ is then made, which usually implies postulating the existence of some coefficient $k$ or $k'$ for which $u_{\mathrm{b}} = k u_{\mathrm{s}}$ or $\overline{u} = k' u_{\mathrm{s}}$ holds true everywhere. Four models participating in ITMIX follow this strategy:

"**Gantayat**" (Gantayat et al., 2014, Supplementary Section S1.5) solves Eq. 4 in elevation bands, and by substituting $\tau$ according to Eq. 3. The central assumption is that $u_{\mathrm{b}} = 0.25\,u_{\mathrm{s}}$. A final, gridded ice thickness distribution is then obtained by smoothing the elevation-band thickness with a 3x3 kernel.

"**RAAJgantayat**" (re-implemented from Gantayat et al., 2014, Supplementary Section S1.14) follows exactly the same procedure. In fact, the method is an independent re-implementation of the "Gantayat" approach.

"**Gantayat-v2**" (adapted from Gantayat et al., 2014, Supplementary Section S1.6), closely follows the original approach by Gantayat et al. (2014). Instead of solving Eq. 4 for elevation bands, however, the equation is solved for discrete points along manually digitized branchlines. Interpolation between various branchlines is then used to obtain an ice thickness distribution. Note that none of the approaches based on the ideas by Gantayat et al. (2014) does account for mass conservation.

"**Rabatel**" (Rabatel et al., unpublished; Supplementary Section S1.16) is based on the knowledge of surface velocities as well, but includes some elements of the mass conserving approaches. Basically, the ice thickness along individual glacier cross sections is calculated by assuming that $\overline{u} = 0.8\,u_{\mathrm{s}}$, and by determining the ice volume flux for a given cross section from an estimate of the mass flux from the upstream area. Combining these informations allows for the area of a given cross section to be computed, and the spatial distribution of $u_{\mathrm{s}}$ along the cross section is used to determine the local ice thickness. The final ice thickness distribution is obtained by interpolation of various cross sections.

## 4.5 Other approaches

This last category includes two additional approaches that cannot be classified in any of the categories above:

"**GCneuralnet**" (Clarke et al., 2009, Supplementary Section S1.8) is based on Artificial Neural Networks (ANN), and thus neglects any kind of glacier physics. The basic assumption is that the bedrock topography underneath glacierized areas closely resembles nearby ice-free landscapes. In principle, the method uses an elevation-dependent azimuthal stencil to "paste" ice-free landscape sections into glacierized parts of a given region.

"**Brinkerhoff**" (Brinkerhoff et al., 2016, Supplementary Section S1.1) poses the problem of finding bedrock elevations in the context of Bayesian inference. The main hypothesis is that both bed elevations and ice flux divergence can be modelled as Gaussian random fields with assumed covariance but unknown mean. Depth-averaged velocities are found by solving the

continuity equation (Eq. 1), and by prescribing normally distributed likelihoods with known covariance around the available velocity, SMB, and $\partial h/\partial t$ data. A Metropolis-Hastings algorithm (Hastings, 1970) is then used to generate samples from the posterior distribution of bed elevations.

## 5   Results and discussion

In total, 189 different solutions were submitted to ITMIX (Tab. 2). Three models ("Farinotti", "Huss", "Linsbauer") were able to handle all 21 test cases, one model handled 20 cases ("Machguth"), and one model handled 19 cases ("Maussion"). Data availability was the main factor hindering the consideration of additional test cases. This is particularly true for the approaches (a) "Brinkerhoff", "Brinkerhoff-v2", "Morlighem", and "VanPeltLeclercq", requiring SMB at least, (b) "Gantayat", "Gantayat-v2", and "RAAJgantayat", requiring surface velocity fields, (c) "Fuerst", requiring SMB, $\partial h/\partial t$ and velocity fields simultaneously and (d) "GCneuralnet", requiring surrounding ice-free terrain for algorithm training. For the approaches "GCbedstress", "RAAJglabtop2", and "Rabatel", the time required for model set up was a deterrent for considering additional test cases.

### 5.1   Between-model comparison

Locally, the solutions provided by the different models can differ considerably. As an example, Figure 2 provides an overview of the solutions generated for the test case "Unteraar" (the real-world case considered by the largest number of models). The large differences between the solutions are particularly evident when comparing the average composite ice thickness (i.e. the distribution obtained when averaging all solutions grid-cell by grid-cell; Fig. 2a) with the local ensemble spread (i.e. the spread between all solutions at a given grid-cell; Fig. 2b). Often, the local spread is larger than the local average. This observation holds true for most of the other test cases as well (not shown).

Figures 2c and 2d provide insights into the composition of the ensemble spread by presenting the composites of the minimum and maximum provided thicknesses, respectively. The models providing the most extreme solutions are depicted in Figures 2e and 2f. In the "Unteraar" example, the approaches "GCneuralnet" and "Fuerst" tend to provide the smallest and largest local ice thickness of the ensemble, respectively. For the specific case, closer inspection shows that the very low ice thicknesses estimated by "GCneuralnet" are associated with the debris covered parts of the glacier, and to the steep slopes delimiting these parts in particular. This is an artefact introduced by the specific setup of the stencil used within the ANN method. In fact, Clarke et al. (2009) found that including steep ice in the definition of valley walls can be advantageous for ANN training. An unforeseen consequence is that steep ice walls close to debris-covered glacier ice are interpreted as valley walls as well, thus causing the surrounding ice thickness to be too thin. Flagging debris-covered glacier parts and treating them as a special case could be an option for alleviating this issue. For "Fuerst", large ice thicknesses (locally exceeding 900 m) mostly occur in the accumulation area. This is the area for which no measured ice flow velocities were available, thus precluding precise model constraint. Uncertainties in this area are propagated downstream thus perturbing the inferred ice thickness even in areas with velocity information. For the particular test case, the approach also provides the minimal ice thickness for large areas, indicating that important oscillations are present in the estimated ice thickness field.

The overall tendency for individual models to provide "extreme" solutions is shown in Figure 3. Two models ("Rabatel" and "GCbedstress") seem to be particularly prone to predict large ice thicknesses, providing the largest ice thickness of the ensemble for 33 and 25 % of the area they considered. Although for "Rabatel" the basis of the statement is weak (only one test case considered) possible explanations lie in (a) the possible overestimation of the area contributing to the ice volume flux of individual profiles, and (b) the assumed relation between depth-averaged and surface flow velocity (cf. Supplementary Sec. S1.16). For "GCbedstress" the possible reasons are less clear. The no-sliding assumption included by the model (see Supplementary Sec. S1.7) – which causes systematically higher thickness estimates than if sliding is assumed – could be a reason. The model, however, seems not to be particularly sensitive to it: Assuming that half of the surface velocity is due to sliding decreases the mean estimated thickness by 13 % only (not shown).

Very small ice thicknesses are often predicted by the models "Maussion" and "GCneuralnet". The two models provided the smallest ice thickness of the ensemble in 30 and 23 % of the considered area, respectively. For "Maussion", the result is mainly driven by the ice thickness predicted for ice caps (Academy, Austfonna, Devon) and large glaciers (Columbia, Elbrus). This is likely related to the applied calibration procedure (cf. Supplementary Sec. S1.12), which is based on data included in GlaThiDa v1. The observations in that dataset, in fact, mostly refer to smaller glaciers (Gärtner-Roer et al., 2014). For "GCneuralnet", it can be noted that the smallest ice thicknesses are often predicted along the glacier centrelines (not shown). Besides the previously discussed issue related to steep ice in proximity of e.g. medial moraines, the ad-hoc solution adopted to allow the ANN stencil to be trained (see Supplementary Sec. S1.8) might be an additional cause.

Although the above observations provide insights into the general behaviour of individual models, it should be noted that a tendency of providing extreme results is not necessarily an indicator of poor model performance. Actual model performance, in fact, can only be assessed through comparison against direct observations (see next section).

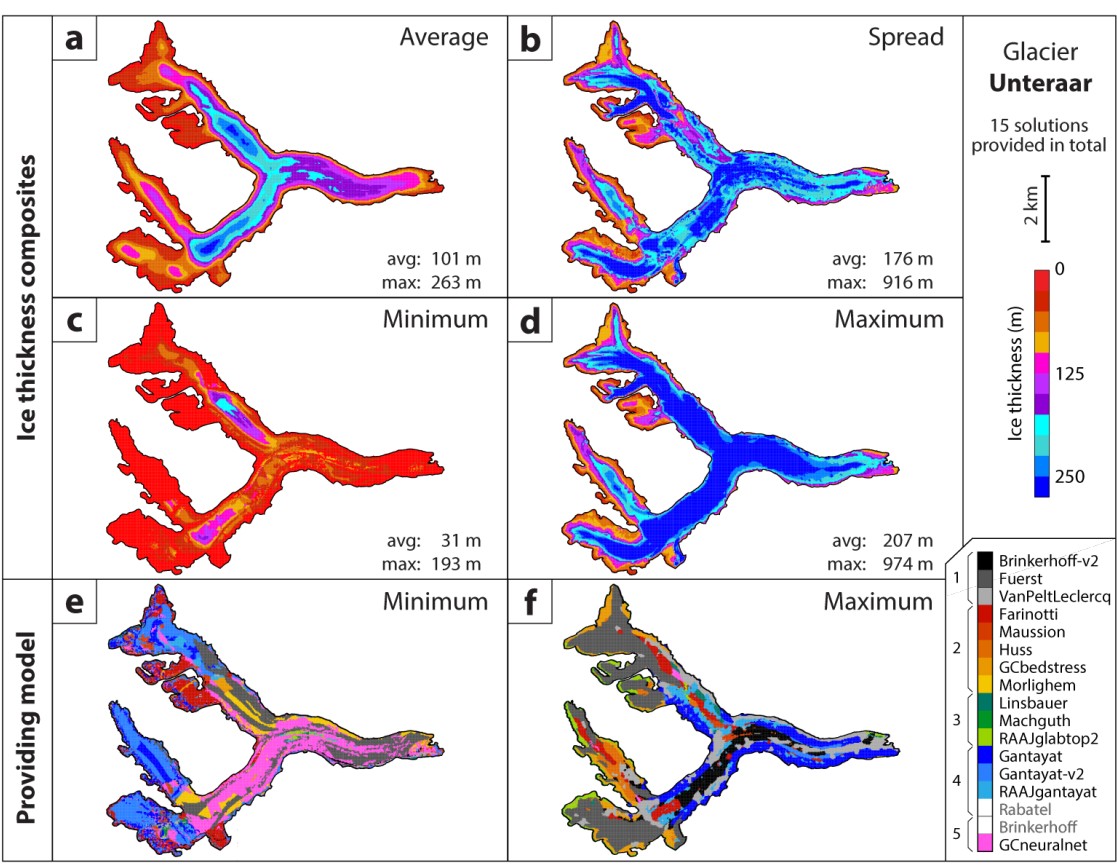

**Figure 2.** Overview of the range of solutions provided by the ensemble of models. The example refers to the test case "Unteraar". The first four panels show composites for the (a) average, (b) spread, (c) minimal, and (d) maximal ice thickness distribution of the 15 submitted solutions. The model providing the minimal and maximal ice thickness for a given location is depicted in panels (e) and (f). Models that did not consider the specific test case are greyed out on the bottom right legend.

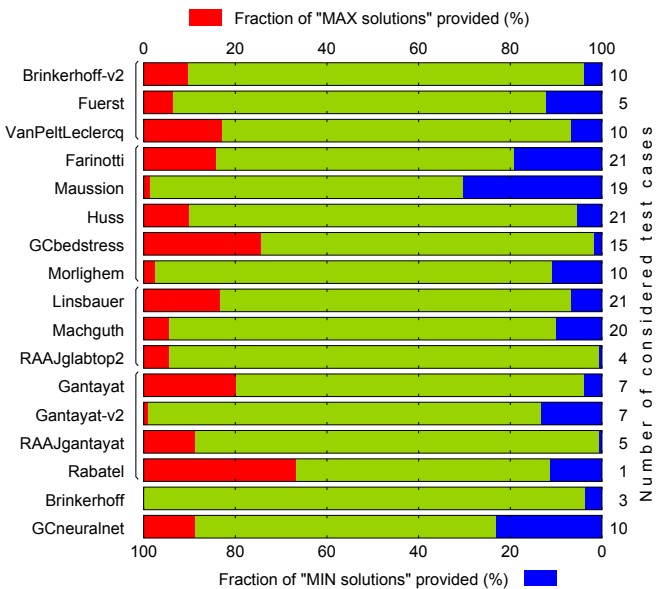

**Figure 3.** Share of "extreme results" provided by individual models. An "extreme result" is defined as either the minimum (MIN) or maximum (MAX) ice thickness occurring in the ensemble of solutions provided for a given test case. The share is based on test case area and assigns equal weights to all cases (a 10 % "fraction of MAX solutions provided" indicates, for example, that on average, the model generated the maximal ice thickness for 10 % of the area of any considered case). The number of test cases considered by individual models is given. Models are sorted according to the categories introduced in Section 4.

## 5.2 Comparison to ice thickness measurements

The solutions submitted by individual models are compared to ice thickness measurements in Figures 4 and 5. For every glacier, the figures show one selected profile along and one across the main ice flow direction. The previously noted large spread between individual solutions re-emerges, as well as the tendency of individual models to produce rather large oscillations. The spread is particularly pronounced for ice caps (Academy, Austfonna, Devon) and for across-flow profiles (Fig. 5).

It is interesting to note that the spread between models is not reduced when individual model categories are considered separately (see also Supplementary Fig. S3). We interpret this as an indication that even models based on the same conceptual principles can be regarded as independent. Whilst this is not surprising for the minimization approaches since they are based on very different forward models (cf. Sec. 4.1), or for the mass conserving approaches since they differ significantly in terms of implementation (Sec. 4.2), the observation is rather unexpected for the shear-stress and velocity based approaches (Sec. 4.3 and 4.4, respectively). The latter two categories, in fact, both rely on very similar concepts. Figure 5 reveals that for shear-stress based approaches the differences are particularly prominent for ice caps, and in the vicinity of ice divides in particular. This seems to be related to the way individual models (a) subdivide individual ice caps, (b) treat the resulting boundaries, and (c) handle very small surface slopes. Also for the participating velocity-based approaches, that apart from "Rabatel" all rely

on the ideas of Gantayat et al. (2014), it seems that the implementation differences of conceptually similar approaches (cf. "Gantayat" and "Gantayat-v2") are sufficient for considering the models as independent.

The above consideration is relevant when interpreting the average solution of the model ensemble (thick green line in Figures 4 and 5): This average solution matches the direct measurements relatively well for most glaciers, with an average deviation below 10 % in 17 out of 21 cases. This increase in prediction accuracy is expected for an unbiased model ensemble. For a set of independent random realizations of the same variable, in fact, Poisson's *law of large numbers* predicts the average result to converge to the expected value (the "true bedrock" in this case) with increasing number of realizations. The so-inferred unbiasedness of the ensemble has an important consequence, as it suggests that future estimates could be significantly improved when relying on such model ensembles. Model weighting – such as used in numerical weather prediction for example (e.g. Raftery et al., 2005) – could additionally be considered in this respect, but would require a sufficiently large data set to quantify model performance.

The positive effect of averaging the results of individual models is best seen in Figure 6. On average over the individual model solutions, the difference between modelled and measured ice thickness is $-17 \pm 36\,\%$ ($1\,\sigma$ estimate) of the mean glacier thickness (first boxplot in the "ALL" group). This value reduces to $+10 \pm 24\,\%$ when the average composite solution is considered, and is close to the value obtained when selecting the best single solution for every test case individually (third and second boxplots of the group, respectively).

Two notable exceptions in the above considerations are given by the test cases "Unteraar" and "Tasman", for which the ensembles of solutions (15 and 11 solutions provided, respectively) converge to a significantly smaller ice thickness than observed (median deviations of $-84\,\%$ and $-65\,\%$, respectively). Two common features that might partially explain the observation are (a) the significant debris cover of the two glaciers, that might bury ice thicker than what would be expected from the present-day SMB fields, and (b) the branched nature of the glaciers, that might be insufficiently captured by the models. Both hypotheses, however, are difficult to test further, as the remaining cases show very different morphological characteristics. An erroneous interpretation of the actual ice thickness measurements, on the other hand, seems unlikely. This is particularly true for Unteraar, for which the reported quality of original radio-echo soundings is high and independent verifications through borehole measurements were performed (Bauder et al., 2003).

"Urumqi" and "Washmawapta", for which 8 and 6 individual solutions were provided respectively, are the other two cases for which the average ice thickness composite differs largely from the observations (median deviations of -71 % and -125 % , respectively; Figs. 4, 5, and 6; recall that because the "true" ice thickness is not known everywhere, deviations are expressed in terms of mean thickness of the average composite). For "Washmawapta" – a cirque glacier mostly fed by steep ice-free headwalls (Sanders et al., 2010) – it is interesting to note that the "Farinotti" approach is the only one predicting ice thickness in the observed range. This suggests that the concept of "ice flow catchments", which is used in the approach for accommodating areas outside the glacier margin that contribute to snow accumulation (cf. Farinotti et al., 2009), is an effective workaround for taking such areas into account. Failure of doing so, in fact, causes the ice volume flux (and thus the ice thickness) to be underestimated. For "Urumqi", on the other hand, the reasons for the substantial underestimation of actual ice thickness are less clear. Potentially, they could be linked to (a) the cold nature of the glacier (e.g Maohuan et al., 1989), which requires thicker

ice to produce a given surface velocity (note that most models assumed flow rate factors for temperate ice; Supplementary Tab. S2), and (b) the artefacts in the provided DEM (note the step-like features in the surface shown in Fig. 4), which lead to locally very high surface slope and thus low ice thickness.

The comparison between Figures 4 and 5 also suggests that, in general, the ice thickness distribution along-flow is better captured than the distribution across-flow. This is likely due to the combination of the fact that most participating approaches include considerations about mass conservation, and that virtually all models include surface slope as a predictor for the local ice thickness. Indeed, these two factors have a stronger control on the along-flow ice thickness distribution than they have across-flow.

The results also indicate that, compared to real-world cases, the ice thickness distribution of the three synthetic cases is better reproduced. On average over individual solutions, the difference to the correct ice thickness is $-17 \pm 20\,\%$ (Fig. 6). This difference reduces to $-15 \pm 11\,\%$ for the average composites, i.e. to a $1\,\sigma$-spread reduced by a factor of two. Again, two factors provide the most likely explanation. On the one hand, the model used for generating the synthetic cases is built upon the same theoretical knowledge as the models used for generating the ice thickness estimates. On the other hand and more importantly, the input data from which the ice thickness distribution is inferred are known without any uncertainty in the synthetic cases. The latter is in contrast to the data available for the real-world cases: Whilst the provided DEMs, $\partial h/\partial t$ fields, and outlines can be considered of good quality, SMB fields are often the product of the inter- and extrapolation of sparse in-situ measurements. The inconsistencies that may arise between $\partial h/\partial t$ and SMB, together with the previously mentioned discontinuities in the available velocity fields (cf. Sec. 3), are obviously problematic for methods that use this information. Two additional observations that might be related to the better model performance in the synthetic cases are (1) that the no-sliding assumption adopted in most models was adequate for the considered synthetic cases, but does not hold true in the real-world ones, and (2) that synthetic glacier geometries were close to steady state. Testing the importance of the second consideration is not possible with the data at hand, and would require the generation of transient synthetic geometries.

In relative terms, the average composite solutions seem to better predict (smaller interquartile range, IQR) the ice thickness distribution of ice caps than that of glaciers. In fact, the $1\,\sigma$-deviations from the measurements for ice caps and glaciers are of $12 \pm 16\,\%$ and $12 \pm 34\,\%$, respectively (Fig. 6). This might be surprising at first, but Figure 4 illustrates that for all three considered ice caps, the average composites are the results of a relatively small set (6 or 7) of largely differing solutions. This issue is particularly evident for the ice cap interiors, for which two model clusters emerge, predicting extremely high and extremely low ice thicknesses, respectively. The relatively small IQR of the ensemble mean, thus, appears to be rather fortuitous, and calls for additional work in this domain. Note, moreover, that the relative accuracy is expressed in relation to the mean ice thickness. In absolute terms, the abovementioned values translate into average deviations in the order of $48 \pm 63\,\text{m}$ for ice caps and $11 \pm 27\,\text{m}$ for glaciers. Obviously, these values are strongly affected by the particular test cases included in the intercomparison, and should not be expected to hold true in general.

To put the average model performance into context, the results are compared to a benchmark model based on volume-area scaling (last boxplot in Fig. 6). The "model" neglects spatial variations in thickness altogether, and simply assigns the mean ice thickness predicted by a scaling relation to the whole glacier. For the scaling relation, we use the form $\overline{h} = cA^{\gamma-1}$, where $\overline{h}$ (m)

and $A$ (km$^2$) are the mean ice thickness and the area of the glacier, respectively. The parameters $c$ and $\gamma$ are set to $c = 0.034$ and $\gamma = 1.36$ for glaciers (Bahr et al., 2015), and to $c = 0.054$ and $\gamma = 1.25$ for ice caps (Radić and Hock, 2010). The values of parameter $\gamma$ have a strong theoretical foundation (Bahr et al., 1997, 2015), whilst $c$ is a free parameter. Since the relation between $c$ and $\overline{h}$ is linear, it must be noted that as long as the distribution of $c$ is symmetric and as long as the value chosen for $c$ corresponds to the mean of that distribution, the results of the above relation correspond to the maximum likelihood estimator for the mean of the distribution of $\overline{h}$. In other words: Randomly sampling different values for $c$ would increase the spread of our estimates, but not its mean.

This simple model deviates from the measured ice thickness by $-42 \pm 59\,\%$, which is a spread (bias) more than twice (four times) as large as estimated for the average composites of the model ensemble ($10 \pm 24\,\%$). This result is reassuring as it suggests that the individual models have actual skill in estimating both the relative ice thickness distribution and the total glacier volume of individual glaciers. The negative sign of the bias – which is consistent with results obtained from a comprehensive dataset in Norway (Andreassen et al., 2015) – should not be overinterpreted, since a different choice for $c$ could be used to alter it. It has again to be noted, however, that this would not reduce the spread in the results, and that for real-world applications, the value of $c$ is unknown. In general, a site-specific calibration of $c$ would be required.

## 5.3 Individual model performance

The considerations in the previous section refer mainly to the average composite ice thickness provided by the ensemble of models. Running a model ensemble, however, can be very impractical. This opens the question on whether individual models can be recommended for particular settings, or whether a single best model can be identified.

To address this question, we propose two separate rankings. Both are based on the (I) average, (II) median, (III) interquartile range, and (IV) 95 % confidence interval (95 % CI) of the distribution of the deviations between modelled and measured ice thicknesses (Fig. 7).

The first ranking considers the individual test cases separately. All models considering a particular test case are first ranked separately for the four indicators (I-IV). When a model does not include a particular test case, no ranks are assigned. For every model, the four indicators are then averaged individually over all test cases. The final rank is computed by computing the mean of these average ranks (Tab. 3). The ranking rewards models with a consistently high performance over a large number of test cases.

The second ranking is only based on the average model performance. In this case, ranks for the above indicators (I-IV) are assigned to the ensemble of point-to-point deviations of the various models (last row of boxplots in Fig. 7; same weight between test cases ensured). The ranks for the four individual indicators are then averaged to obtain the overall rank (Tab. 4). In contrast to the first option, this ranking does not consider the test cases individually, and does not account for the number of considered test cases. A model considering only one test case but performing perfectly on it, for example, would score highest.

The ranking result of every model on a case-by-case basis is given in Supplementary Table S3. The distributions of the deviations between modelled and measured ice thicknesses for every model and considered test case are given in Figure 7 and

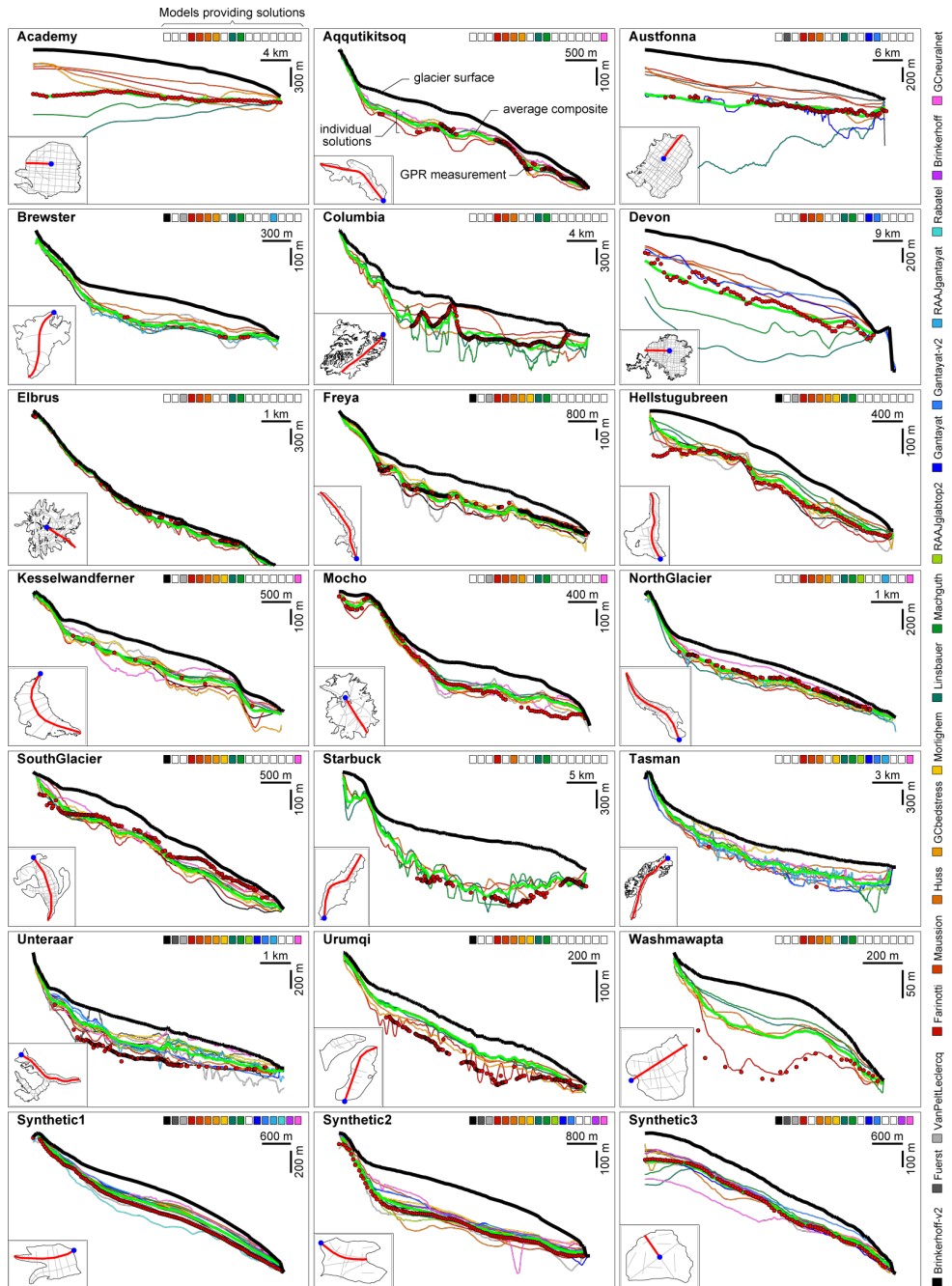

**Figure 4.** Comparison between estimated and measured bedrock topographies. For every test case, a longitudinal profile showing the glacier surface (thick black line), the bedrock solution of individual models (coloured lines), the average composite solution (thick green line), and the available GPR measurements (black-encircled red dots) are given. The coloured squares on the upper left of the panels indicate which models provided solutions for the considered test case (see legend on the right margin for colour key). The location of the profiles are shown on the small map on the bottom left of the panels (red), and the beginning of the profile (blue dot) is to the left. Available ice thickness measurements are shown in grey.

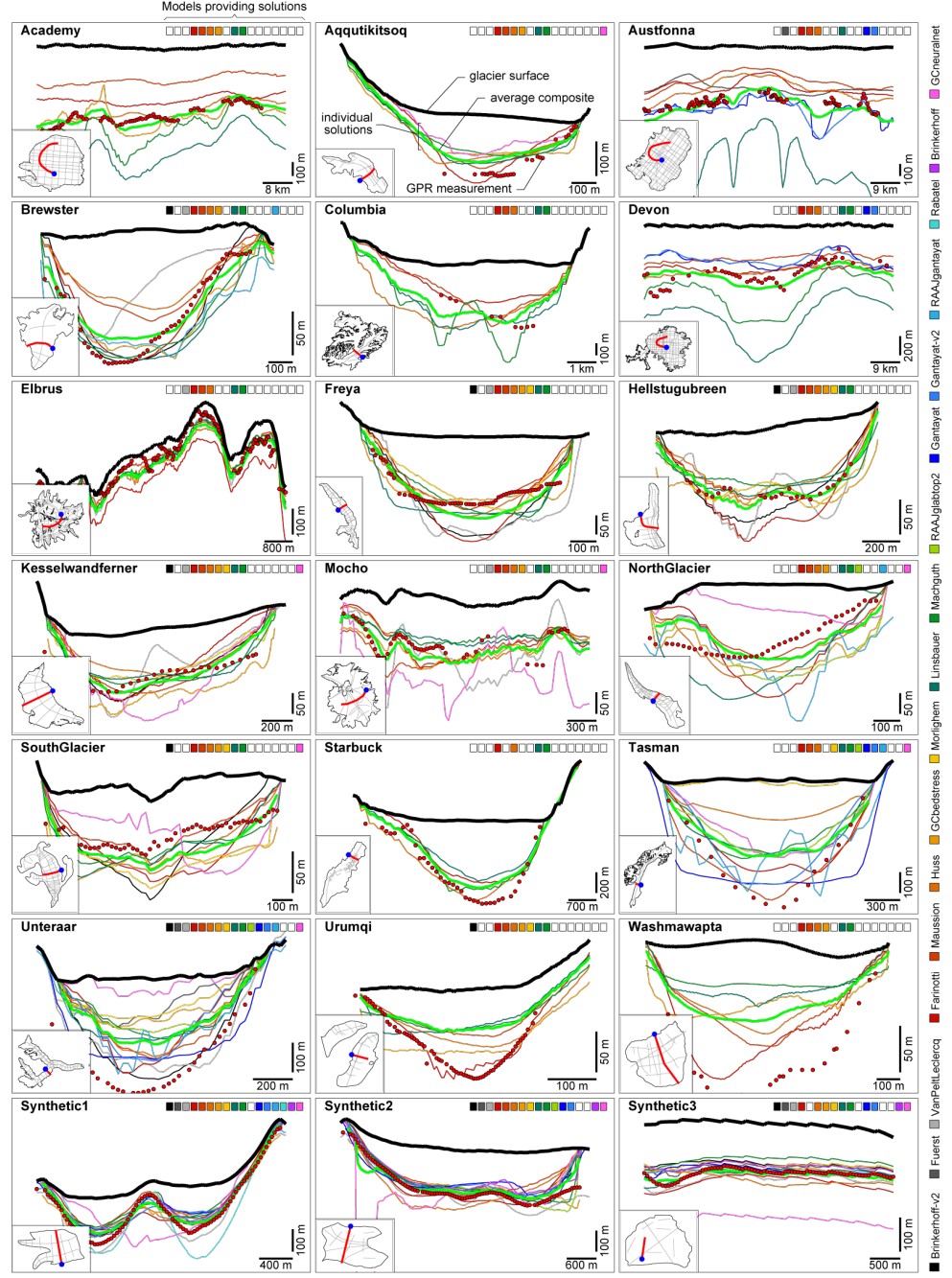

**Figure 5.** Same as Figure 4, for a series of cross-sectional profiles.

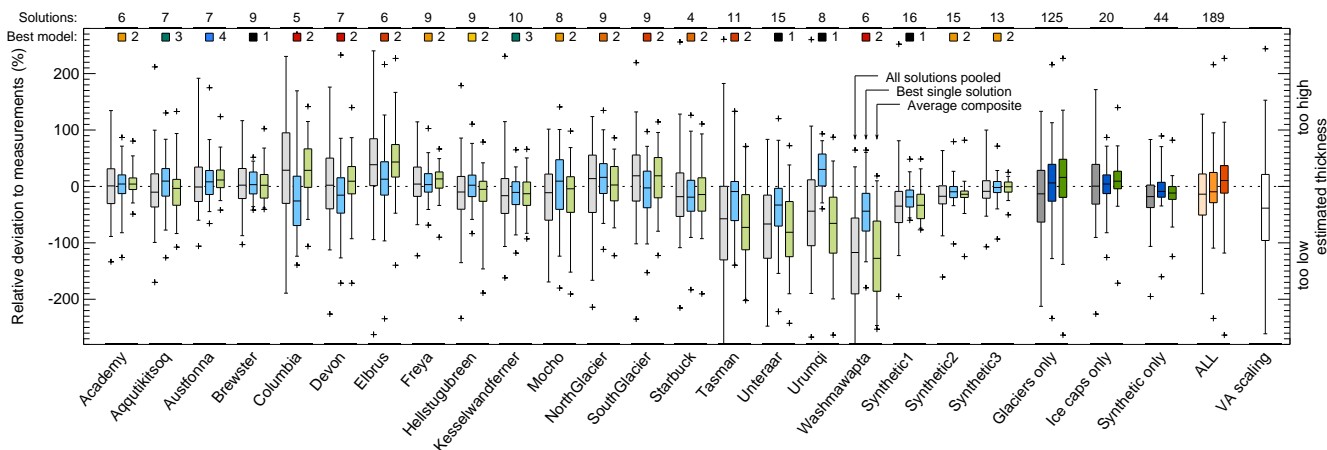

**Figure 6.** Effect of merging individual model solutions. For every test case, the distribution of the deviations between modelled and measured ice thicknesses is shown for the case in which (i) the individual point-to-point comparisons of all available solutions are pooled (grey boxplots), (ii) only the provided single best solution is considered (blue boxplots), and (iii) the deviations are computed from the average composite thickness of all model solutions (green boxplots). Deviations are expressed relative to the mean ice thickness. The best single solution is computed by summing the ranks for the (a) average deviation, (b) median deviation, (c) interquartile range, and (d) 95 % confidence interval. The distributions of the deviations when grouping glaciers, ice caps, and synthetic glaciers separately are additionally shown, as are the results when grouping all test cases together (ALL). When forming the groups, point-to-point deviations for every test case are resampled so that every test case has the same weight. The last boxplot to the right refers to the case in which the mean ice thickness is predicted by volume-area scaling (see Section 5.2). The upper part of the panel provides the number of considered model solutions, and the model providing the single best solution (see Fig. 4 for colour key; the number refers to the model category according to Section 4). Boxplots show minimum and maximum values (crosses), the 95 % confidence interval (whiskers), the interquartile range (box) and the median (lines within box).

Supplementary Figure S2. Similarly as noted during the discussion of the last section (Sec. 5.2), the rankings do not suggest a performance advantage in any of the five model categories introduced in Section 4.

Combined over the two rankings, the model "Brinkerhoff-v2" scores highest (3nd and 1st rank, respectively). The good score is mainly driven by the comparatively small model spread (IQR and 95 % CI) and bias (Tab. 4). The small model bias (−3 % average deviation), however, arises from a partial compensation between positive bias for glaciers (+5 %) and negative bias for the synthetic cases (−22 %) (Tab. 4). Unfortunately, the model did not consider any ice cap, thus hampering any statement on model performance in this particular setting. Ice caps were not considered mainly because of the absence of the necessary data.

Apart from the model "Brinkerhoff-v2", the first positions in the first ranking are occupied by models that consider a large number of test cases (Tab. 3). The model by "Maussion" is rated highest. Similar to "Brinkerhoff-v2", the good result is driven by the small IQRs and 95 % CIs, in particular for glaciers and ice caps. In the second ranking, the model is severely penalised (11th rank) for its large bias (−36 % on average; Tab. 4). The bias is particularly prominent in the case of ice caps and the

425 synthetic cases (−42 and −45 % average deviation, respectively), and may be related to the fact that the "Maussion" model was developed and calibrated by using data from valley glaciers only. For the synthetic cases in particular, the calibration with real-world glaciers (i.e. cases that include sliding) seems to be a likely explanation for a systematic underestimation of the ice thickness. This, however, appears to be only a partial explanation, as such a negative bias is apparent for most approaches, i.e. also for approaches that explicitly assumed no sliding (e.g. "GCbedstress", "Morlighem"; cf. Supplementary Sec. S1).

In general, the model bias can be interpreted as an indicator for the performance of the models in reproducing the total glacier ice volume. The latter is not discussed explicitly as the computation of a "measured volume" would need the available measurements to be interpolated over large distances. Seven of the considered models show a bias of less than 8 % (Tab. 4). An interesting case in this respect is given by the model by Gantayat et al. (2014), which yields small biases (−4 and −8 %) for both considered implementations ("Gantayat" and "RAAJgantayat", respectively). The relatively low overall ranks assigned to

these models (ranks 10 and 14 in the first ranking, ranks 3 and 12 in the second, respectively) are an expression of the relatively small number of considered test cases (first ranking), and the relatively large model spread (second ranking). Of interest is also the observation that the version of the model considering multiple flowlines ("Gantayat-v2") yields a significant higher bias (−32 % on average) than the approach based on elevation bands, despite a moderate decrease in model spread. The increase is particularly visible for real-world glaciers, for which the bias changes from +4 % to −61 %. This might hint at the difficulty in

correctly subdividing a given glaciers into individual flowlines, and could be an indication that the rather mechanistic procedure used in this case (cf. Supplementary Sec. S1.6) is insufficient for achieving a sensible subdivision.

The difficulty in correctly interpreting the overall model bias is well illustrated in the case of the "Linsbauer" model: The model yields the smallest bias over the entire set of considered test cases (−1 % on average), but is the result of a compensation between (a) a moderate negative bias for glaciers and the synthetic test cases (both −16 %), and (b) a large positive bias for ice

caps (+91 %).

Together with "Brinkerhoff-v2", the model "Farinotti" is the second one included in the first five places of both rankings (ranks 4 and 5, respectively; Tabs. 3 and 4). The relatively high ranking is due to a combination of comparatively high model performance (small bias and spread) and large number of considered test cases. The consideration of all test cases, however, should not be interpreted as capability of handling large samples of glaciers in this case. The application of the model, in fact,

requires a significant amount of manual input (cf. Sec. 4). This is in contrast to the fully automated methods of "Maussion", "Huss", and "Machguth". In this respect it is interesting to note that the model by "Huss" slips from the $2^{\text{th}}$ rank in the first ranking to the $8^{\text{th}}$ in the second one. The relatively low score in the second ranking is mainly an expression of the comparatively large confidence intervals (Tab. 4). Combined over the two rankings, however, the model can be considered as the best amongst the fully automated approaches.

The model "GCbedstress" ($5^{\text{th}}$ and $6^{\text{th}}$ in the two rankings) ranks highest when only ice caps are considered. The average deviation of $3 \pm 17$ % indeed suggests a very high model performance. However, it has to be noted that the result is based on one test case only (Tab. 2). For the models considering more ice caps, the results are heterogeneous and difficult to interpret, as models showing small IQRs show large bias, and vice versa (Tab. 4).

The model "GCneuralnet" is found at the other end of the ranking (penultimate and last ranks, respectively). The average deviation of $-39 \pm 52\%$ highlights both the large bias and large spread of the estimates. Obviously, the performance of approaches based on ANN are highly dependent on the data set used for algorithm training. The large deviations might therefore be an expression of the issues encountered with the provided DEMs (cf. Supplementary Sec. S1.8), rather than an indication of generally low model performance. As already noted, however, the general absence of ice-free analogues for ice caps or crater glaciers makes the approach unsuitable for this kind of morphologies.

An interesting result emerges when considering the IQRs and 95 % CIs in the synthetic test cases (cf. Tab. 4): Approaches that include SMB, $\partial h/\partial t$ or velocity information in addition to the glacier outline and the DEM of the surface (e.g. approaches by "Brinkerhoff", "Fuerst", "Morlighem", or "VanPeltLeclerq") yield the smallest spreads around the average deviation (IQR < 22 % for all mentioned models). This is in marked contrast to the real-world cases, in which the IQRs are about five times larger (average IQR for the same models = 105 %), and similar considerations apply when analysing the 95 % CIs. As already noted, this is most likely linked to the differences in data quality. Whilst the input data for the synthetic cases are perfectly known, the data available for the real-world cases were necessarily retrieved from various, independent data sources (cf. Tab. 1). This often led to problems in the mutual consistency of the surface fields, and caused particular difficulties to those models requiring all of the information. The capability of accounting for observational uncertainties, as in the "Brinkerhoff" approach for example (cf. Supplementary Sec. S1.1), hence seems to be an important prerequisite when handling real-world cases. Similarly, having access to reliable uncertainty estimates for any particular dataset would be important. Emphasis has to be put in this domain if significant advances are to be achieved.

## 6 Conclusions

ITMIX was the first coordinated intercomparison of approaches that estimate the ice thickness of glacier and ice caps from surface characteristics. The goal was to assess model performance for cases in which no a-priori information about ice thickness is available. The experiment included 15 glaciers and 3 ice caps spread across a range of different climatic regions, as well as 3 synthetically generated test cases.

ITMIX attracted 13 research groups with 17 different models that can be classified into (1) minimization approaches, (2) mass conserving approaches, (3) shear-stress based approaches, (4) velocity-based approaches, and (5) other approaches outside of the previous categories. The 189 solutions submitted in total provided insights into the performance of the various models, and the accuracies that can be expected from their application.

The submitted results highlighted the large deviations between individual solutions, and even between solutions of the same model category. The local spread often exceeded the local ice thickness. Caution is thus required when interpreting the results of individual models, especially if they are applied to individual sites. Substantial improvements in terms of accuracy, however, could be achieved when combining the results of different models. Locally, the mean deviation between an average composite solution and the measured ice thickness was in the order of $10 \pm 24\%$ of the mean ice thickness ($1\sigma$ estimate). This hints at the random nature of individual model errors, and suggest that ensembles of models could help in improving the estimates. For

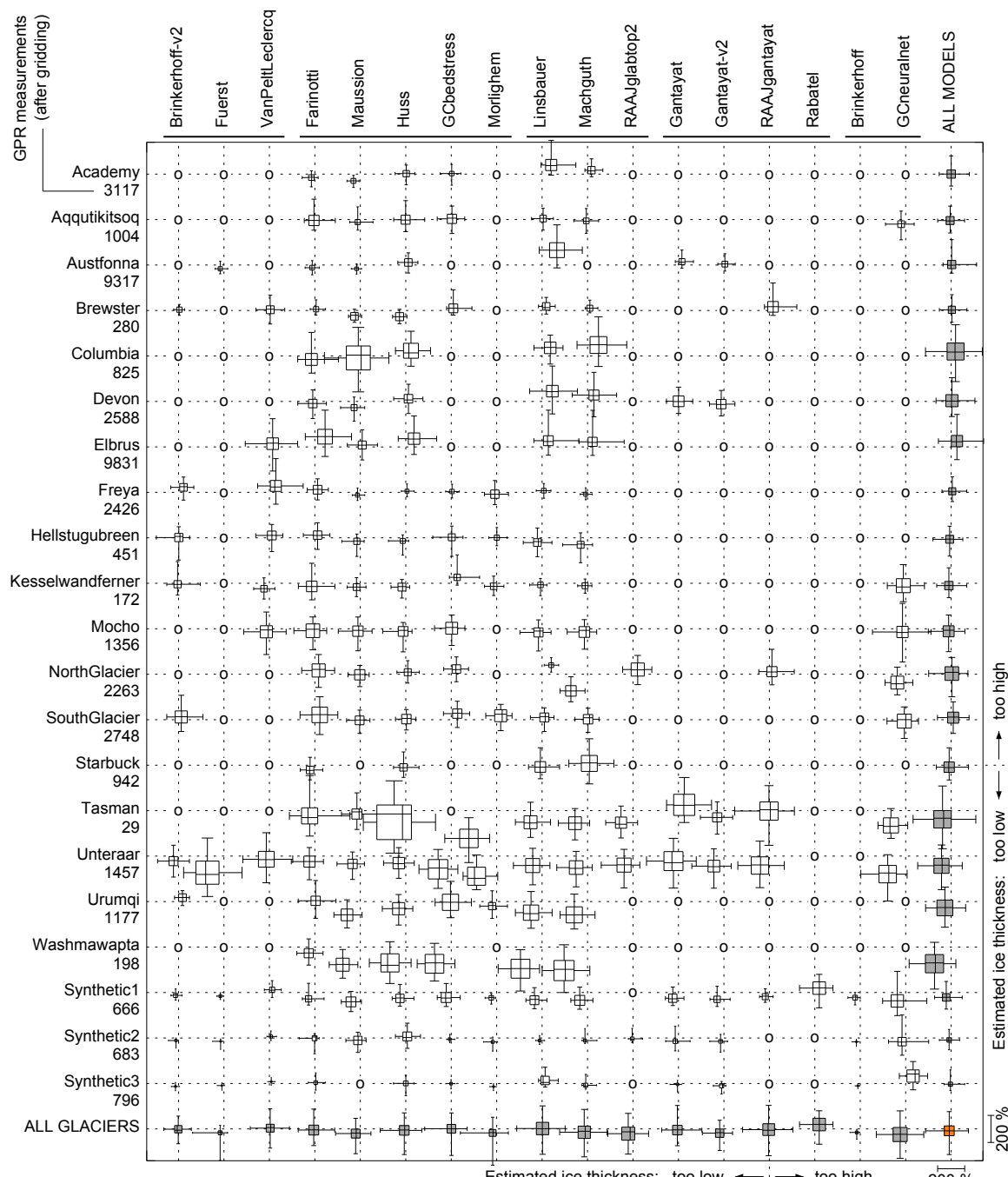

**Figure 7.** Difference between estimated and measured ice thicknesses. For every test case (rows) and every model (columns; ordered according to the categories defined in Sec. 4), the distribution of the point-by-point deviations between estimated and measured ice thicknesses is shown. Differences are expressed relative to the mean ice thickness (e.g., a 100 % deviation means that the modelled ice thickness deviates from the measured one by one mean ice thickness). Circles indicate that no solution was submitted. Boxplots show the 95 % confidence interval (whiskers), the interquartile range (box) and the median (lines within box). The boxplots are squared to facilitate the comparison within models and within test cases. Note the scale bars in the bottom right corner.

**Table 3.** Ranking of individual models based on case-by-case performance. Numbers in front of model names refer to the categories introduced in Section 4. Displayed values are mean ranks for the average (avg), median (med), interquartile range (IQR) and 95 % confidence interval (95 %) of the deviations from ice thickness measurements. For every group (glaciers, ice caps, and synthetic cases), the average of the above three ranks is given (AVG). Values for "ALL" correspond to the average of the three groups. The three best average results for every group are given in bold. $n$ is the number of considered test cases per model.

| Model - version | n | Glaciers only | | | | | Ice caps only | | | | | Synthetic only | | | | | ALL | | | | |
|---|---|---|---|---|---|---|---|---|---|---|---|---|---|---|---|---|---|---|---|---|---|
| | | avg | med | IQR | 95% | AVG | avg | med | IQR | 95% | AVG | avg | med | IQR | 95% | AVG | avg | med | IQR | 95% | AVG |
| 2 Maussion | 19 | 4.6 | 4.3 | 2.9 | 3.0 | **3.7** | 5.3 | 5.3 | 1.3 | 1.0 | 3.2 | 13.0 | 11.0 | 12.5 | 10.5 | 11.8 | 5.6 | 5.2 | 3.7 | 3.5 | **4.5** |
| 2 Huss | 21 | 4.5 | 4.3 | 4.3 | 3.5 | **4.1** | 2.0 | 2.0 | 4.3 | 5.0 | 3.3 | 4.7 | 3.3 | 12.0 | 12.7 | 8.2 | 4.1 | 3.8 | 5.4 | 5.0 | **4.6** |
| 1 Brinkerhoff-v2 | 10 | 3.6 | 3.6 | 4.6 | 5.3 | **4.2** | - | - | - | - | - | 7.7 | 6.7 | 3.3 | 4.3 | 5.5 | 4.8 | 4.5 | 4.2 | 5.0 | **4.6** |
| 2 Farinotti | 21 | 3.7 | 3.9 | 5.5 | 5.6 | 4.7 | 2.7 | 3.3 | 2.7 | 3.0 | **2.9** | 5.3 | 6.0 | 9.0 | 10.7 | 7.8 | 3.8 | 4.1 | 5.6 | 6.0 | 4.9 |
| 2 GCbedstress | 15 | 4.6 | 4.8 | 5.7 | 5.5 | 5.2 | 2.0 | 1.0 | 1.0 | 5.0 | **2.2** | 4.0 | 5.3 | 8.3 | 7.3 | 6.2 | 4.3 | 4.7 | 5.9 | 5.9 | 5.2 |
| 1 VanPeltLeclercq | 10 | 4.1 | 4.3 | 7.1 | 7.9 | 5.9 | - | - | - | - | - | 4.3 | 4.0 | 5.3 | 4.7 | **4.6** | 4.2 | 4.2 | 6.6 | 6.9 | 5.5 |
| 3 Linsbauer | 21 | 5.4 | 5.3 | 3.5 | 3.9 | 4.5 | 6.7 | 6.7 | 6.7 | 6.7 | 6.7 | 11.0 | 8.3 | 9.0 | 8.0 | 9.1 | 6.4 | 6.0 | 4.8 | 4.9 | 5.5 |
| 3 Machguth | 20 | 5.9 | 5.6 | 4.1 | 3.7 | 4.8 | 4.5 | 4.5 | 5.0 | 4.5 | 4.6 | 8.7 | 10.7 | 10.0 | 10.3 | 9.9 | 6.2 | 6.2 | 5.0 | 4.8 | 5.5 |
| 5 Brinkerhoff | 3 | - | - | - | - | - | - | - | - | - | - | 9.7 | 9.3 | 2.3 | 2.0 | 5.8 | 9.7 | 9.3 | 2.3 | 2.0 | 5.8 |
| 4 RAAJgantayat | 5 | 4.8 | 4.8 | 7.8 | 9.8 | 6.8 | - | - | - | - | - | 4.0 | 3.0 | 6.0 | 5.0 | **4.5** | 4.6 | 4.4 | 7.4 | 8.8 | 6.3 |
| 3 RAAJglabtop2 | 4 | 7.0 | 6.7 | 6.7 | 6.3 | 6.7 | - | - | - | - | - | 1.0 | 3.0 | 9.0 | 8.0 | 5.2 | 5.5 | 5.8 | 7.2 | 6.8 | 6.3 |
| 1 Fuerst | 5 | 13.0 | 14.0 | 15.0 | 15.0 | 14.2 | 4.0 | 6.0 | 2.0 | 2.0 | 3.5 | 7.7 | 8.0 | 1.3 | 3.3 | **5.1** | 8.0 | 8.8 | 4.2 | 5.4 | 6.6 |
| 2 Morlighem | 10 | 6.7 | 6.7 | 6.0 | 4.9 | 6.1 | - | - | - | - | - | 12.0 | 11.3 | 4.3 | 3.7 | 7.8 | 8.3 | 8.1 | 5.5 | 4.5 | 6.6 |
| 4 Gantayat | 7 | 4.0 | 4.0 | 11.0 | 10.0 | 7.2 | 3.5 | 2.0 | 5.5 | 4.0 | 3.8 | 8.0 | 7.7 | 8.7 | 9.0 | 8.3 | 5.6 | 5.0 | 8.4 | 7.9 | 6.7 |
| 4 Gantayat-v2 | 7 | 7.0 | 8.0 | 3.0 | 6.5 | 6.1 | 2.5 | 2.5 | 4.0 | 3.0 | **3.0** | 11.0 | 10.0 | 9.3 | 8.7 | 9.8 | 7.4 | 7.3 | 6.0 | 6.4 | 6.8 |
| 5 GCneuralnet | 10 | 7.1 | 7.9 | 7.3 | 6.9 | 7.3 | - | - | - | - | - | 9.7 | 14.0 | 14.0 | 14.7 | 13.1 | 7.9 | 9.7 | 9.3 | 9.2 | 9.0 |
| 4 Rabatel | 1 | - | - | - | - | - | - | - | - | - | - | 5.0 | 5.0 | 16.0 | 15.0 | 10.2 | 5.0 | 5.0 | 16.0 | 15.0 | 10.2 |

applications at the large scale – such as the estimation of the ice thickness distribution of an entire mountain range and beyond – reducing the uncertainties through such a strategy will be challenging, as only few models are currently capable of operating at the regional or global scale.

Although no clear pattern emerged for the performance of individual model categories, the intercomparison allowed statements about the performance of individual models. The model "Brinkerhoff-v2" was detected as the best single model, with average deviations for real-world glaciers in the order of $-3 \pm 27\,\%$. Some caution has to be expressed, however, since the model considered only about half of the provided test cases and was not applied to any ice cap. The model "Huss" scored highest amongst the automated methods capable of handling large sets of glaciers. With average deviations of $-14 \pm 35\,\%$, the approach ranged mid-way when considering point-to-point deviations from measurements. For ice caps, the model "GCbedstress" showed very promising results (average deviations of $3 \pm 17\,\%$), although generalizing this observation would be speculative, as the approach considered only one test case. For ice caps, particularly large differences between individual models were detected in the proximity of ice divides. This calls for improvements in how models treat these regions.

**Table 4.** Ranking of individual models based on average model performance. Numbers in front of model names refer to the categories introduced in Section 4. Displayed values are the average (avg), median (med), interquartile range (IQR) and 95 % confidence interval (95 %) of the percental deviations from ice thickness measurements. The three best results for every column is given in bold. $n$ is the number of considered test cases per model. $AVG$ is the average of the ranks assigned to the values in the three "ALL" columns.

| Model - version | n | Glaciers only avg | med | IQR | 95% | Ice caps only avg | med | IQR | 95% | Synthetic only avg | med | IQR | 95% | ALL avg | med | IQR | 95% | AVG |
|---|---|---|---|---|---|---|---|---|---|---|---|---|---|---|---|---|---|---|
| 1 Brinkerhoff-v2 | 10 | 5 | 6 | ±**33** | ±**120** | - | - | - | - | −22 | −19 | ±10 | ±39 | −3 | −9 | ±27 | ±**102** | 3.8 |
| 5 Brinkerhoff | 3 | - | - | - | - | - | - | - | - | −29 | −26 | ±10 | ±36 | −29 | −26 | ±**10** | ±**36** | 5.8 |
| 4 Gantayat | 7 | 4 | −7 | ±72 | ±200 | 16 | 14 | ±29 | ±93 | −22 | −18 | ±22 | ±68 | −4 | −7 | ±35 | ±148 | 6.0 |
| 1 VanPeltLeclercq | 10 | **2** | 4 | ±43 | ±168 | - | - | - | - | 15 | 13 | ±11 | ±47 | 6 | 9 | ±29 | ±145 | 6.0 |
| 2 Farinotti | 21 | **1** | −3 | ±45 | ±148 | −21 | −21 | ±22 | ±**76** | −13 | −3 | ±27 | ±76 | −4 | −7 | ±36 | ±135 | 6.5 |
| 2 GCbedstress | 15 | −4 | 7 | ±**39** | ±168 | 3 | 4 | ±17 | ±77 | −15 | −9 | ±17 | ±71 | −6 | **1** | ±32 | ±157 | 6.5 |
| 3 Linsbauer | 21 | −16 | −4 | ±46 | ±167 | 91 | 79 | ±48 | ±159 | −16 | −17 | ±20 | ±106 | −1 | **2** | ±46 | ±180 | 8.0 |
| 2 Huss | 21 | −21 | −13 | ±39 | ±171 | **13** | **11** | ±26 | ±90 | **−9** | −10 | ±27 | ±89 | −14 | −8 | ±35 | ±154 | 8.5 |
| 4 Gantayat-v2 | 7 | −61 | −62 | ±41 | ±**140** | −5 | −3 | ±29 | ±90 | −30 | −25 | ±22 | ±64 | −32 | −28 | ±31 | ±**112** | 8.8 |
| 1 Fuerst | 5 | −113 | −135 | ±86 | ±228 | −26 | −30 | ±**16** | ±**47** | −24 | −22 | ±**9** | ±**28** | −42 | −26 | ±**14** | ±131 | 9.0 |
| 2 Maussion | 19 | −34 | −26 | ±**36** | ±**142** | −42 | −39 | ±**19** | ±**60** | −45 | −43 | ±35 | ±89 | −36 | −31 | ±33 | ±131 | 10.5 |
| 4 RAAJgantayat | 5 | **−3** | **3** | ±50 | ±201 | - | - | - | - | −28 | −27 | ±22 | ±50 | −8 | −10 | ±43 | ±186 | 10.8 |
| 3 Machguth | 20 | −34 | −23 | ±52 | ±190 | 39 | 33 | ±33 | ±132 | −26 | −24 | ±15 | ±91 | −26 | −18 | ±45 | ±175 | 11.0 |
| 4 Rabatel | 1 | - | - | - | - | - | - | - | - | 29 | 34 | ±46 | ±123 | 29 | 34 | ±46 | ±123 | 11.5 |
| 3 RAAJglabtop2 | 4 | −42 | −43 | ±62 | ±162 | - | - | - | - | **−2** | −9 | ±13 | ±53 | −32 | −21 | ±48 | ±151 | 12.2 |
| 2 Morlighem | 10 | −55 | −26 | ±48 | ±237 | - | - | - | - | −32 | −28 | ±11 | ±41 | −47 | −27 | ±**26** | ±215 | 12.5 |
| 5 GCneuralnet | 10 | −55 | −50 | ±53 | ±181 | - | - | - | - | **0** | −15 | ±63 | ±158 | −39 | −40 | ±52 | ±176 | 15.8 |

Somewhat surprisingly, models that include SMB, $\partial h/\partial t$ or surface flow velocity fields in addition to the glacier outline and DEM did not perform better when compared to approaches requiring less data, in particular for real-world cases. Inconsistencies between available datasets – which are often acquired with very different techniques, spatial footprints, and temporal resolutions – appeared to be the most likely cause. Although it must be noted that the set of considered synthetic cases was generated upon the same theoretical knowledge as the approaches used for ice thickness inversion, the generally better model performance for these cases supports the previous hypothesis. In the synthetic cases, in fact, input data were known precisely, i.e. without observational errors. This highlights the importance for mutually consistent data sets, and suggests that improved observational capabilities could help to improve the performance of the next generation of ice thickness estimation methods. Similarly, improving the model's capability of taking into account uncertainties in the input data should be considered a priority.

Besides improved data concerning glacier surface characteristics, a key for developing a new generation of ice thickness estimation models will be the data base against which the models can be calibrated and validated. The data utilized within ITMIX are available as a supplement to this paper (see link at the end of this section), but a much larger effort is ongoing in collaboration with the World Glacier Monitoring Service. With the initiation of the Glacier Thickness Database (Gärtner-Roer et al., 2014; WGMS, 2016), the first steps towards a freely accessible, global database of ice thickness measurements have

been undertaken. We anticipate that this effort, together with a second phase of ITMIX targeting at how to best integrate sparse thickness measurements to improve model performance, will foster the development of improved ice thickness estimation approaches.

To summarize: In order to improve available thickness estimates for glacier and ice caps, we make the following recommendations:

- Ensemble-methods comprising a variety of independent, physically-based approaches should be considered. This is likely to be a more effective strategy than focusing on one individual approach.
- Models should be extended to take observational uncertainty into account. The Bayesian framework used by Brinkerhoff et al. (2016), for example, showed promising results in this respect.
- The increasing availability of surface ice-flow velocity data (e.g. Scambos et al., 2016) should be exploited. In this context, the previously-mentioned necessity of accounting for observational uncertainty is crucial.
- The way individual models treat ice divides has to be improved. This is important when addressing ice caps and glacier complexes, and to ensure consistency between subsurface topographies of adjacent ice masses.
- Efforts for centralizing available ice thickness measurements should be strengthened. Initiatives such as the GlaThiDa database launched by the World Glacier Monitoring Service are essential for new generations of ice thickness models to be developed and validated.

All data used within ITMIX, as well as the individual solutions submitted by the participating models, can be found at

<code>http://3cm.ch/ITMIX_full_dataset.zip</code>

*Author contributions.* DF and HL designed ITMIX. DF coordinated the experiment, performed the model evaluations, prepared the figures, and wrote the paper with contributions by DB, GKCC, JF, PG, FG-C, MH, PWL, AL, HM, FM, MM, AR, RR, OS, and WJJvP. DB, GKCC, JJF, HF, PG, FG-C, CG, MH, PWL, AL, HM, CMa, FM, MM, CMo, APa, APo, AR, RR, OS, SSK, and WJJvP participated in the experiment providing modelling results. BA, LMA, TB, DB, JAD, DF, AF, KH, SK, IL, HL, RMcN, and PAS provided the data necessary for the real-world test cases. CMa and DF generated the synthetic test cases. MH, LMA and GHG provided advice during experimental setup and evaluation.

*Acknowledgements.* ITMIX was made possible by the International Association of Cryospheric Sciences (IACS). We are indebted to the European Space Agency's project *Glaciers_cci*, as well as to Elisa Bjerre, Emiliano Cimoli, Gwenn Flowers, Marco Marcer, Geir Moholdt, Johnny Sanders, Marius Schäfer, Konrad Schindler, Tazio Strozzi, and Christoph Vogel for providing data necessary for the experiment. DF acknowledges support from the Swiss National Science Foundation (SNSF). MM and CG were funded by the National Aeronautics and Space Administration, Cryospheric Sciences Program, grant NNX15AD55G. PWL was funded by the European Research Council under the European Union's Seventh Framework Programme (FP/2007–2013)/ERC Grant Agreement No. 320816. JAD acknowledges the UK Natural Environment Research Council for supporting the airborne radio-echo sounding surveys of the three Arctic ice caps included in the experiment (grants GR3/4663, GR3/9958, GR3/12469 and NE/K004999/1). AR and OS acknowledge the contribution of Labex

OSUG@2020 (Investissements d'avenir - ANR10 LABX56) and the VIP_Mont-Blanc project (ANR-14-CE03-0006-03). FG-C and CMo acknowledge support from French National Research Agency (ANR) under the SUMER (Blanc SIMI 6) 2012 project ANR-12BS06-0018. The constructive comments by Shin Sugiyama and an anonymous referee helped to improve the manuscript.

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
