# Peer review of "How accurate are estimates of glacier ice thickness?"

_The Cryosphere, 2016_

## Referee Comment (RC1) · S. Sugiyama (Referee) · 21 Jan 2017

General comments:

This paper presents a comparison of models which compute thickness of glaciers and ice caps from surface observations. Ice thickness estimation from surface information is a classical problem in glaciology, but it is still very important and unsolved issue. Several algorithms have existed from earlier time, and more approaches and applications have been proposed and tested for the last 10 years. Such recent development benefits from growing amount of satellite derived data (e.g. DEMs and velocity map). Because increasing number of studies are presented in this field, I find it very important and timely to call an experiment to compare performance of existing models.

[Figure]

The authors designed an interesting experiment. The selected 18 glaciers have a large variety of geographical and geometrical settings, which enabled the authors to analyze the robustness and weakness of each model. In addition to the real existing glaciers, synthetically generated glaciers are included in the test cases, which is useful to study the physics behind the models. The authors also made a fairly good job in presenting a large amount of data produced by 17 different models. I particularly acknowledge their effort to use the ice thickness data as much as possible to evaluate and rank the model results. Ranking models is tricky, but this paper follows a quantitative criteria, and the descriptions are careful and modest.

This kind of model comparison is useful for researchers in the field because it helps to (1) evaluate the accuracy of each model and (2) to find direction to improve the models. Moreover, it is beneficial for researcher not exactly in the field because it provides (3) insight into the robustness of the modelling in general and (4) overview of the modelling approaches and techniques. In my opinion, the paper is well done in terms of (1)-(3) and a little more effort is needed to improve (4). Otherwise, the paper is well constructed and carefully written. I list several points, hoping the paper becomes more readable and useful for readers not in the field of ice thickness modeling.

1. Classification of the models The models are classified into four approaches (1. mass-conservation only, 2. mass-conservation and momentum equation, 3. shallow ice approximation, 4. artificial neural networks) (line 109-112). However, the descriptions on the models are not grouped nor ordered according to the physics behind. Moreover, no detailed explanations are given for these four approaches, and the model descriptions are mostly on technical issues. This makes it difficult to distinguish the models, and follow the interpretations on the model comparison results. I suggest the authors to elaborate on Section 4, so that physical basis and fundamental equations of the four approaches are given, and model descriptions are ordered and structured in terms of the approach. Once basic information and equations are given, each model description can refer the information to explain how they solve the problem and what is

the difference from other models using the same approach.

2. "unpublished" models Several models are "unpublished". Because details are not explained, results from these models are difficult to evaluate and increase uncertainties in the overall discussion. If there is no written material available as a reference, details of the models should be provided in Supplementary Material. Without knowledge of "how they computed the model", it is not much useful to include the results into the comparison and statistical treatment.

3. Averaging the model results One of the conclusions of the paper is "accuracy of the ice thickness estimation improves by averaging results of different models" (e.g. line 5-7, 396-400 and 554-556). Given the large variations over the models, I found this argument too simplistic. It would be OK, for example, if all the models rely on the same physics and only detailed treatments are different. In this case, models are classified into four groups and the number of models in each group are different. I wonder if the same argument stands if models belong to one of the groups are averaged. Otherwise, does it mean all of the approaches are equally wrong but to different directions? To get into this point, classification of the models and understanding of the difference between the models are important. I hope to see more logical interpretation on why the accuracy improves after averaging.

4. Rate factor A I understand that each model used their own value for the rate factor A. Since this is a crucial parameter for ice thickness estimation, I suggest the author to indicate the value for each of the models. I am also concerned about the use of constant A for cold and temperate glaciers. "Huss" applies a temperature dependent A (line 233), but not clear for the others. I also wonder if this value is taken as a tuning factor in some models. Can you provide more information when you describe the model approaches in Section 4?

Specific comments:

line 9: "sensitivity to input data consistency" » I understood this only after reading the

manuscript. Please consider to rewrite.

line 34: "... inverse modelling of ice thickness from glacier ice flow ..."?

line 35: "additional properties" » What exactly? Basal slipperiness? something else?

line 38: "a rapid pace:" » "a rapid pace."?

line 70: Can you describe more about the sample glaciers and ice caps? For example, how many calving glaciers? Mean, maximum and minimum glacier length and area? Temperate of cold? Can you also describe here, why ice sheets are excluded from the experiment?

line 82: "Consistent glacier-wide estimates ..." » What exactly do you mean by "consistent"?

line 84: "separate tiles" » Not very clear

line 94: "accumulation and ablation zone" » "accumulation and ablation zones"?

line 98: "suitable size and shape" » Do you mean the model did not reach a steady state?

line 124: "non-physical behavior incompatible with ..." » What do you mean?

line 140: I suppose the rate factor A is a very important parameter for estimating ice thickness from velocity. I was a little surprised to see different value was used in each model. Are they tuned in each model? I suggest to indicate the values used for A for all the models.

line 141: "effective mass balance" » What exactly is this?

line 249: "designed for alpine glaciers" » What would be a problem for ice caps? Too flat surface geometry? How did you compute thickness when surface slope is zero?

line 315: "vertical gradients" » What is this?

line 333: "constant forcing" » Do you mean constant SMB?

line 340-344: "Requirement of input data" is a fundamental feature of the models. This should be explained in the previous section and used to classify the models.

line 343-344: "the average composite ice thickness (i.e. the composite of the local average thickness computed from the ensemble of provided solutions)" » Is this the average of all provided thickness for a certain grid point? It is not clear and confusing.

line 344-3435: "local ensemble spread" » Is this a technical term? If not, please define more precisely. Something like "range of computed ice thickness variation at each grid point"?

line 371: "possible reason are" » "possible reason is"

line 392-394: I wonder if the ice thicknesses computed by the models are independent because the models are classified into several approaches.

line 405: "branched nature" » If this was the reason, larger deviations are expected near the confluence area, but it appears not.

line 453-462: I am a little surprised at this relatively short discussion on the comparison with the volume-area scaling method. Since volume-area scaling is commonly used for large scale ice volume estimation, its accuracy and reliability is a big concern of the readers. Isn't it useful to elaborate more on this issue, for example, by changing the parameters c and gamma?

line 463: "average composite solution provided by the ensemble of models" » This is not very clear expression.

line 551: "The relative low..." » "The relatively low..."

line 572: "data were known precisely" » In my opinion, the synthetic glaciers were better reproduced by the models because the glacier thickness is distributed as expected from our knowledge and this knowledge is used in the models. In this regard, "precisely

known" may not be a primary reason why the performance was better in the synthetic cases. I agree that accurate and more complete data sets are important for the models, but the logic here is a little problematic.

Figure 7: This plot is interesting, but very difficult to read useful information. I understand that this plot can be replaced by a set of plots showing the results of a model applied for all the glaciers (Figures S1 and S2), and another set of plots showing the results for a glacier computed by all the models. If you use Figure 7 in the paper, can you provide the latter set of plots in Supplementary Material?

---

## Referee Comment (RC2) · Anonymous Referee #2 · 7 Feb 2017

Inventory of glaciers on Earth has been recently achieved, however ice thickness distribution remains largely unknown at a global scale for obvious reasons of tremendous geophysical survey it would require to be completed. A comprehensive knowledge of glaciers thickness is however of primary importance for a number of glaciological studies: estimating the volume of ice stored (water availability, sea-level change) and a prerequisite before any ice flow-modeling attempt. This work presents the results of the first intercomparison of the currently used methodologies to estimate thickness of a glacier from surface measurements. This community effort is highly relevant for at least two reasons. First, as mentioned, improving our knowledge of the thickness of glaciers would open to significant progresses in fields of high societal impacts. Second, the amount of surface data available will most likely significantly increase in the forthcoming years and appropriate methodologies to infer ice thickness distribution will have to be developed. The paper is well structured and written; I therefore recommend its publication in the best delay.

I must confess that I had one slight frustration reading the manuscript: there is not that many recommendations on further direction of developments. I clearly have in mind how difficult this could be in such a large community effort with a lot of variety in the approaches used and the incomplete realization of the whole set of experiments by most of the models participating. I have one suggestion to circumvent this difficulty, which I believe would also improve the readability of the paper and facilitate its impact on non-specialists. 17 different models have been used, and according to the authors, they could be classified in 4 categories. In the manuscript the description and discussion are only viewed from a single model point of view or from the entire set of models. I think that approaching also the manuscript using these 4 categories would strengthen the paper. Some suggestions regarding that point:

- Section 4, is rather technical and probably gives not enough details for specialists and too many for non-specialists. I would recommend arranging section 4 in order to present the philosophy of each type/category of method and to add in the supplementary materials required details for each model. Regarding that latter aspect, I would strongly encouraged unpublished approaches to give more details as information are today to my opinion very limited.

- In more or less all figures and tables (and particularly in table1, table 2, table 4, Figure 3, Figure 7) the 17 models are presented in alphabetic order. I would recommend grouping them along the 4 categories. Color code of figures 4, 5 might also be re-arranged to try to give the opportunity to the reader of disentangling the approach used behind the models.

- Are they pattern emerging regarding these categories? Systematic bias for a given

category for some specific configuration (glaciers, ice caps, synthetic)? In other words is there a family of approach that is more suited for a given configuration or not? I think this should be discussed. My guess is that if there were some obvious ones it would have been already discussed. However, mentioning that today we are not able to discriminate whether one family of approach seems more appropriate than another is also a result that I think deserve to be mentioned.

I would further encourage having a perennial repository of all the datasets required for people to confront future approaches with the results already computed (i.e. set up and results). This is mentioned as an intention in the conclusion, but a clear link to an already established database would be much better. Ideally, this database should be reviewed with the manuscript.

---

## Author Comment (AC1) · 3 Mar 2017

**TC-2016-250**
**Authors final response**

Daniel Farinotti, and the ITMIX consortium

We would like to thank the referees and the editor for the time invested in considering our manuscript, and for the positive and constructive feedback.

In the following, each reviewer comment (RC) is listed together with our author response (AR). Where applicable, the proposal for a revised version of the text appears below in smaller script size and quotation marks.

**Comments by Reviewer #1 (Shin Sugiyama)**

**RC:** *This paper presents a comparison of models which compute thickness of glaciers and ice caps from surface observations. Ice thickness estimation from surface information is a classical problem in glaciology, but it is still very important and unsolved issue. Several algorithms have existed from earlier time, and more approaches and applications have been proposed and tested for the last 10 years. Such recent development benefits from growing amount of satellite derived data (e.g. DEMs and velocity map). Because increasing number of studies are presented in this field, I find it very important and timely to call an experiment to compare performance of existing models.*

*The authors designed an interesting experiment. The selected 18 glaciers have a large variety of geographical and geometrical settings, which enabled the authors to analyze the robustness and weakness of each model. In addition to the real existing glaciers, synthetically generated glaciers are included in the test cases, which is useful to study the physics behind the models. The authors also made a fairly good job in presenting a large amount of data produced by 17 different models. I particularly acknowledge their effort to use the ice thickness data as much as possible to evaluate and rank the model results. Ranking models is tricky, but this paper follows a quantitative criteria, and the descriptions are careful and modest.*

*This kind of model comparison is useful for researchers in the field because it helps to (1) evaluate the accuracy of each model and (2) to find direction to improve the models. Moreover, it is beneficial for researcher not exactly in the field because it provides (3) insight into the robustness of the modelling in general and (4) overview of the modelling approaches and techniques. In my opinion, the paper is well done in terms of (1)-(3) and a little more effort is needed to improve (4). Otherwise, the paper is well constructed and carefully written. I list several points, hoping the paper becomes more readable and useful for readers not in the field of ice thickness modeling.*

**AR:** We thank the reviewer for his very positive appreciation and do value his recommendation regarding the overview of the various methods. Since the same point is also at the heart of the comments by Reviewer #2 (see below), we changed the manuscript structure in this respect. Instead of providing a short model-by-model description, we now classify the individual models in five different categories and only provide a description of the principles behind each category.

For an extended, more in-depth description of the individual models, we now refer the reader either to the supplementary material (for unpublished approaches) or to the original publications (for approaches that have previously been published). More detailed information on how we implemented these changes are presented hereafter, when replying to the individual reviewer's comments.

———

**RC:** *1. Classification of the models: The models are classified into four approaches (1. mass-conservation only, 2. mass-conservation and momentum equation, 3. shallow ice approximation, 4. artificial neural networks) (line 109-112). However, the descriptions on the models are not grouped nor ordered according to the physics behind. Moreover, no detailed explanations are given for these four approaches, and the model descriptions are mostly on technical issues. This makes it difficult to distinguish the models, and follow the interpretations on the model comparison results. I suggest the authors to elaborate on Section 4, so that physical basis and fundamental equations of the four approaches are given, and model descriptions are ordered and structured in terms of the approach. Once basic information and equations are given, each model description can refer the information to explain how they solve the problem and what is the difference from other models using the same approach.*

**AR:** We thank the reviewer for this constructive comment that we implemented by re-designing *S*ection 4. Instead of presenting the individual models one-by-one and in alphabetical order, we now have five sub-sections describing the principles behind different model categories. Our classification is now based on five categories that include (1) approaches casting ice thickness inversion as a minimization problem, (2) approaches based on mass conservation, (3) approaches based on a parametrization of basal shear stress, (4) approaches based on observed surface velocities, and (5) other approaches not belonging to any of the previous categories. Each sub-section describing one of these categories is split into a part presenting the principles upon which a given category is based, and a part in which we briefly highlight how individual model-implementations differ from each other. This should not only better clarify differences between the individual models, but also facilitate the overall readability of the manuscript, since this solution allowed us to significantly shorten the corresponding section.

———

**RC:** *2. "unpublished" models: Several models are "unpublished". Because details are not explained, results from these models are difficult to evaluate and increase uncertainties in the overall discussion. If there is no written material available as a reference, details of the models should be provided in Supplementary Material. Without knowledge of "how they computed the model", it is not much useful to include the results into the comparison and statistical treatment.*

**AR:** We absolutely agree with the reviewer's comment and followed his suggestion: A more detailed description of the unpublished models is now found in the Supplementary (see Supplementary Section S1).

———

**RC:** *3. Averaging the model results: One of the conclusions of the paper is "accuracy" of the ice thickness estimation improves by averaging results of different models" (e.g. line 5-7, 396-400 and 554-556). Given the large variations over the models, I found this argument too simplistic. It would be OK, for example, if all the models rely on the same physics and only detailed treatments are different. In this case, models are classified into four groups and the number of models in each group are different. I wonder if the same argument stands if models belong to one of the groups are averaged. Otherwise, does it mean all of the approaches are equally wrong*

*but to different directions? To get into this point, classification of the models and understanding of the difference between the models are important. I hope to see more logical interpretation on why the accuracy improves after averaging.*

**AR:** In this case, we are of a slightly different opinion as the reviewer. Our central argument is that the "improvement through averaging" is observed because of the independence of individual models. This independence includes, amongst other, the reliance on different methodological principles (we are somewhat reluctant in calling this "different physics", as we think that this wording is too strong). Stated differently: We would not expect the improvement to be so effective if only models based on the same approach were to be averaged. In this sense, and very loosely speaking, yes, one could say that "all approaches are equally wrong" – with some of them being less wrong than others.

Trying to detect which category of approaches provides the best results when considered by its own is obviously the final goal of the intercomparison. Unfortunately, no clear picture emerges from our analyses, which is both reflected in the various figures and in the two rankings that we propose. One obstacle in achieving more robust results is the fact that not all models considered (or were able to consider) all test cases. As shown in Table 2, and explained at the beginning of Section 5, data availability was the major obstacle in having individual models considering more test cases.

In the revised manuscript, we tried to better address the above aspects. Also following the suggestion by Reviewer #2, we better clarified that no clear picture emerges when analysing the results in categories. We emphasized this by introducing a new paragraph at the beginning of Section 5, by rewording the section that the reviewer refers to, and by presenting a new figure in the Supplementary (Supplementary Fig. S3). In the latter, we evaluate the overall model performance for individual model categories separately. We hope this to satisfactorily address the reviewer's concern, and in particular to clarify our logics behind the "averaging" argument.

**Section 5.2, Paragraph 2**

"It is interesting to note that the spread between models is not reduced when individual model categories are considered separately (see also Supplementary Fig. S3). We interpret this as an indication that even models based on the same conceptual principles can be regarded as independent. Whilst this is not surprising for the minimization approaches since they are based on very different forward models (cf. Sec. 4.1), or for the mass conserving approaches since they differ significantly in terms of implementation (Sec. 4.2), the observation is rather unexpected for the shear-stress and velocity based approaches (Sec. 4.3 and 4.4, respectively). The latter two categories, in fact, both rely on very similar concepts. Figure 5 reveals that for shear-stress based approaches the differences are particularly prominent for ice caps, and in the vicinity of ice divides in particular. This seems to be related to the way individual models (a) subdivide individual ice caps, (b) treat the resulting boundaries, and (c) handle very small surface slopes. Also for the participating velocity-based approaches, that apart from "Rabatel" all rely on the ideas of Gantayat et al. (2014), it seems that the implementation differences of conceptionally similar approaches (cf. "Gantayat" and "Gantayat-v2") are sufficient for considering the models as independent.

The above consideration is relevant when interpreting the average solution of the model ensemble (thick green line in Figures 4 and 5): This average solution matches the direct measurements relatively well for most glaciers, with an average deviation below 10 % in 17 out of 21 cases. This increase in prediction accuracy is expected for an unbiased model ensemble. For a set of independent random realizations of the same variable, in fact, Poisson's *law of large numbers* predicts the average result to converge to the expected value (the "true bedrock" in this case) with increasing number of realizations. The so-inferred unbiasedness of the ensemble has an important consequence, as it suggests that future estimates could be significantly improved when relying on such model ensembles. Model weighting – such as used in numerical weather prediction for example (e.g. Raftery et al., 2005)

– could additionally be considered in this respect, but would require a sufficiently large data set to quantify model performance."

————

**RC:** *4. R̲ate factor A: I understand that each model used their own value for the rate factor A. Since this is a crucial parameter for ice thickness estimation, I suggest the author to indicate the value for each of the models. I am also concerned about the use of constant A for cold and temperate glaciers. "Huss" applies a temperature dependent A (line 233), but not clear for the others. I also wonder if this value is taken as a tuning factor in some models. Can you provide more information when you describe the model approaches in Section 4?*

**AR:** With the values chosen for the rate factor $A$, the reviewer raises an important point. Indeed, there are differences in how individual models treat this parameter. One important issue to note is that no information about ice temperature was provided within the experiment – nor is it available in general. To date, only few ice thickness estimation models have a scheme for estimating $A$ from available (meteo/climatic) information, and – as correctly noted by the reviewer – the value of the parameter is most commonly prescribed or tuned.

We address all these aspects by introducing Supplementary Table S2, which now lists (a) the numerical value (or range of values) used for $A$, (b) a flag for whether the parameter is kept constant for all test cases or adapted, and (c) a flag for whether the parameter is prescribed or used as a tuning factor.

————

**RC:** *line 9: "sensitivity to input data consistency" >> I understood this only after reading the manuscript. Please consider to rewrite.*

**AR:** The sentence now reads:

**Abstract**
" Models relying on multiple data sets – such as surface ice velocity fields, surface mass balance, or rates of ice thickness change – showed high sensitivity to input data quality."

————

**RC:** *line 34: "...inverse modelling of ice thickness from glacier ice flow..."?*
*line 35: "additional properties" >> What exactly? Basal slipperiness? something else?*

**AR:** Reworded into:

**Section 1, Paragraph 4**
"Alternative methods based on more rigorous inverse-modelling, on the other hand, have often focused on additionally inferring basal slipperiness together with bedrock topography [...]"

————

**RC:** *line 38: "a rapid pace:" >> "a rapid pace."?*

**AR:** Corrected as suggested.

————

**RC:** *line 70: Can you describe more about the sample glaciers and ice caps? For example, how many calving glaciers? Mean, maximum and minimum glacier length and area? Temperate of cold? Can you also describe here, why ice sheets are excluded from the experiment?*

**AR:** We are of the opinion that stating mean, minimum, and maximum values in the text

wouldn't be particularly well-readable. We added the information about calving in Table 1. Since reliable information about the thermal state (temperate/polythermal/cold) is not available for all test cases, we preferred not to include it. A short explanation for why ice sheets were excluded is provided.

**Section 3, Paragraph 1**

"Since most published approaches for estimating ice thickness were developed for applications on mountain glaciers and smaller ice caps, ice sheets where not included in the experiment."

————

**RC:** *line 82: "Consistent glacier-wide estimates..." >> What exactly do you mean by "consistent"?*

**AR:** We meant "estimates that rely on the same methodology and that have a homogeneous data quality across the glacier". For the sake of simplicity, we removed the adjective. The sentence now reads:

**Section 3, Paragraph 3**

Glacier-wide estimates of surface velocities were not available for any of the considered cases.

————

**RC:** *line 84: "separate tiles" >> Not very clear*

**AR:** Changed to "separate sources"

————

**RC:** *line 94: "accumulation and ablation zone" >> "accumulation and ablation zones"?*

**AR:** We believe that the singular form is correct (a glacier has not multiple ablation zones).

————

**RC:** *line 98: "suitable size and shape" >> Do you mean the model did not reach a steady state?*

**AR:** Strictly speaking, no. As stated in the sentence that follows (former line 99) "[the] resulting geometries were *c*lose to steady state". In the data accompanying the paper (link to dataset) we provide distributed fields of ice-thickness change rates for all of the synthetic test cases, so that the interested reader can quantify what change rate corresponds to "close" in this context. We don't see an added value in mentioning maximal or average change rates values in the text.

————

**RC:** *line 124: "non-physical behaviour incompatible with..." >> What do you mean?*

**AR:** By this we meant that some combinations of provided SMB and dh/dt fields implied upwards flow velocities, or similar. The entire model description was revised and moved to the Supplementary in response to the reviewer's "general comments". The wording is no longer found.

————

**RC:** *line 140: I suppose the rate factor A is a very important parameter for estimating ice thickness from velocity. I was a little surprised to see different value was used in each model. Are they tuned in each model? I suggest to indicate the values used for A for all the models.*

**AR:** See reply to "general comment 4": The answers are now given in Supplementary Table S3.

————

**RC:** *line 141: "effective mass balance" >> What exactly is this?*

**AR:** The wording is introduced by the "Brinkerhoff" model to describe the term $\dot{b} - \frac{\partial S}{\partial t}|_{\text{obs}}$, where $\dot{b}$ is the climatic mass balance, and $\partial t|_{\text{obs}}$ the observed ice thickness change rate. The section was moved to the Supplementary (see Section S1.2) and the wording is now properly introduced.

————————

**RC:** *line 249: "designed for alpine glaciers" >> What would be a problem for ice caps? Too flat surface geometry? How did you compute thickness when surface slope is zero?*

**AR**: The problem is that such methods are conceived by having in mind individual, constrained ice bodies that show one main flow direction (typically some valley-type glacier). For applying such methods to ice caps, the latter need first to be subdivided into individual sections referring to independent ice-flow basins. It is true that such a subdivision is performed within the Randolph Glacier Inventory, but still, the resulting elements have morphological characteristics that are very different from typical valley-type glaciers. Also the consistency of the ice thickness at ice divides is currently addressed insufficiently by models developed for mountain glaciers.

For what the question regarding "zero surface slope" is concerned, most models apply some sort of filtering (i.e. they impose a minimal surface slope).

The first question should no longer arise with the revision of Section 4, whilst the latter aspect is now addressed in Section 4.2:

**Section 4.2, following Eq. (2)**

"[...] $\alpha$ is surface slope [...] To avoid infinite $h$ for $\alpha$ tending to zero, a minimal surface slope is often imposed, or $\alpha$ is averaged over a given distance. Based on theoretical considerations (Kamb and Echelmeyer, 1986), this distance should correspond to 10-20 times the ice thickness."

————————

**RC:** *line 315: "vertical gradients" >> What is this?*

**AR:** Here we referred to the rate with which SMB varies with elevation, i.e. $\partial b/\partial z$. This is clarified in the revised methodology description found in Supplementary Section S1.16.

————————

**RC: line 333: "constant forcing" >> Do you mean constant SMB?**

**AR:** Correct. This is clarified in the revised description (Supplementary Section S1.17).

————————

**RC:** *line 340-344: "Requirement of input data" is a fundamental feature of the models. This should be explained in the previous section and used to classify the models.*

**AR:** We partly disagree with the reviewer, in the sense that different approaches allow for an optional use of a particular data set. We agree, however, that individual approaches are characterized by the need of a particular data set. The first observation is addressed in the right-hand side of Table 2, whilst we better clarified the second observation in the revised Section 4. Details about the particular data requirements for individual models are, moreover, found in the extended model descriptions of the Supplementary (Supplementary Section S1).

————————

**RC:** *line 343-344: "the average composite ice thickness (i.e. the composite of the local average*

*thickness computed from the ensemble of provided solutions)" >> Is this the average of all provided thickness for a certain grid point? It is not clear and confusing.*

**AR:** Well, yes, but for all grid cells (therefore the wording "composite"). We changed the wording as follows, and hope this to be clearer:

**Section 5.1, Paragraph 1**

"[...] the average composite ice thickness (i.e. the distribution obtained when averaging all solutions grid-cell by grid-cell [...])."

————————

**RC:** *line 344-3435: "local ensemble spread" >> Is this a technical term? If not, please define more precisely. Something like "range of computed ice thickness variation at each grid point"?*

**AR:** We introduced the following clarification:

**Section 5.1, Paragraph 1**

"[...] the local ensemble spread (i.e. the spread between individual solutions at a given grid-cell [...])."

————————

**RC:** *line 371: "possible reason are" >> "possible reason is"*

**AR:** "possible reasons are"

————————

**RC:** *line 392-394: I wonder if the ice thicknesses computed by the models are independent because the models are classified into several approaches.*

**AR:** This is not a trivial question, and would basically require the introduction of a metric for the degree of (in-)dependence of various models. Provocatively, one may even ask whether in earth sciences independent models exist at all. Some fundamental principles, in fact, are probably included in every kind of model (take "gravity" as an extreme example...). In our opinion, the fact that individual models can be grouped into categories does not compromises their independence for the analyses we perform. Indeed, even for conceptually identical models (cf. "Machguth" and "RAAJglabtop2"), the results are that far apart from each other that a dependence is questionable. We therefore argue that independence can be claimed also for models belonging to the same group.

————————

**RC:** *line 405: "branched nature" >> If this was the reason, larger deviations are expected near the confluence area, but it appears not.*

**AR:** We are not sure from where "it appears not". When inspecting Figure 2b (which shows the spread between individual model solutions), we indeed have the impression that the zones with highest spread seem to cluster before and after confluence areas. We agree that this hypothesis is not particularly strong, but we are of the opinion that this is adequately communicated in the sentence starting on the same line ("Both hypotheses [...] are difficult to test further, as the remaining cases show very different morphological characteristics "). As far as the reviewer has no strong objection, we would like to keep the hypothesis mentioned in the text.

————————

**RC:** *line 453-462: I am a little surprised at this relatively short discussion on the comparison with the volume-area scaling method. Since volume-area scaling is commonly used for large scale*

*ice volume estimation, its accuracy and reliability is a big concern of the readers. Isn't it useful to elaborate more on this issue, for example, by changing the parameters c and gamma?*

**AR:** The reviewer is absolutely correct in noting that different values for the scaling parameters (factor $c$ and exponent $\gamma$) would yield to different results. And it is true that the issue could be addressed by randomly sampling such parameters from a suitable population (this is what we understand the reviewer is suggesting). It has to be noted, however, that the value of $\gamma$ has a strong theoretical foundation (cf. Bahr et al., 2015), and that only $c$ should therefore be considered as a random variable (Bahr et al., 2015). Since $c$ has a linear influence on mean ice thickness $\overline{h}$ (it is $V = cA^{\gamma}$, and thus $\overline{h} = V/A = cA^{\gamma-1}$), randomly sampling $c$ from a symmetric distribution, would then only change the width of the distribution of the result, but not its mean. In this sense, as long as the value chosen for $c$ represents the mean value of the distribution of $c$, our results can be interpreted as the maximum likelihood estimator for the resulting $\overline{h}$. Or stated otherwise: As long as $c$ corresponds to the mean value of its distribution, the spread that Fig. 6 shows for the scaling approach is the minimal possible spread that one needs to expect from scaling. Any other parameter choice would enlarge this spread without changing the position around which the spread occurs.

We tried to better explain this line of reasoning by adding the following clarification:

**Section 5.2, penultimate and last Paragraph**

"The values of parameter $\gamma$ have a strong theoretical foundation (Bahr et al., 1997, 2015), whilst $c$ is a free parameter. Since the relation between $c$ and $\overline{h}$ is linear, it must be noted that as long as the distribution of $c$ is symmetric and as long as the value chosen for $c$ corresponds to the mean of that distribution, the results of the above relation correspond to the maximum likelihood estimator for the mean of the distribution of $\overline{h}$. In other words: Randomly sampling different values for $c$ would increase the spread of our estimates, but not its mean. [...] The negative sign of the bias – which is consistent with results obtained from a comprehensive dataset in Norway (**?**) – should not be overinterpreted, since a different choice for $c$ could be used to alter it. It has again to be noted, however, that this would not reduce the spread in the results, and that for real-world applications, the value of $c$ is unknown. In general, a site-specific calibration of $c$ would be required.
* * *
**RC:** *line 463: "average composite solution provided by the ensemble of models" >> This is not very clear expression.*

**AR:** We changed "average composite solution" into "average composite ice thickness", which is the wording we defined previously (see also the reviewer's comment for Lines 343-344).
* * *
**RC:** *line 551: "The relative low..." >> "The relatively low..."*

**AR:** Changed as suggested.
* * *
**RC:** *line 572: "data were known precisely" >> In my opinion, the synthetic glaciers were better reproduced by the models because the glacier thickness is distributed as expected from our knowledge and this knowledge is used in the models. In this regard, "precisely known" may not be a primary reason why the performance was better in the synthetic cases. I agree that accurate and more complete data sets are important for the models, but the logic here is a little problematic.*

**AR**: We agree with the reviewer that the better performance for the synthetic test cases may

in part be due to the fact that the model used for generating the cases is built upon the same knowledge as the models used for ice thickness reconstruction. And in fact, we do acknowledge this in the text (former lines 430-431: *"[...] two factors provide the most likely explanation. On the one hand, the model used for generating the synthetic cases is built upon the same theoretical knowledge as the models used for generating the ice thickness estimates. On the other [...]"*). The comment by the reviewer, however, relates to a different context. In that part, we observed that the models one would consider to be more sophisticated because of the capacity of integrating various sources of information, did not necessarily perform better in a real-world setting than more simple models (i.e. models requiring less input data). Our suggestion is that this may indeed be related to data quality issues since in the synthetic cases – where data are known without uncertainties – these models did indeed perform well. It in this sense that we do not fully understand what part of our logic would be "problematic".

————

**RC:** *Figure 7: This plot is interesting, but very difficult to read useful information. I understand that this plot can be replaced by a set of plots showing the results of a model applied for all the glaciers (Figures S1 and S2), and another set of plots showing the results for a glacier computed by all the models. If you use Figure 7 in the paper, can you provide the latter set of plots in Supplementary Material?*

**AR:** The reviewer's request would generate 22 additional figures, and we think that this would not be beneficial for maintaining an overview. The data for producing such figures are provided to the reader (link to dataset), but we would like to avoid including this large amount of additional items in the Supplementary. Figure 7 is thus maintained, but adjusted to reflect the model categorization suggested by both the reviewers.

**Comments by Reviewer #2 (Anonymous)**

**RC:** *Inventory of glaciers on Earth has been recently achieved, however ice thickness distribution remains largely unknown at a global scale for obvious reasons of tremendous geophysical survey it would require to be completed. A comprehensive knowledge of glaciers thickness is however of primary importance for a number of glaciological studies: estimating the volume of ice stored (water availability, sea-level change) and a prerequisite before any ice flow-modeling attempt. This work presents the results of the first intercomparison of the currently used methodologies to estimate thickness of a glacier from surface measurements. This community effort is highly relevant for at least two reasons. First, as mentioned, improving our knowledge of the thickness of glaciers would open to significant progresses in fields of high societal impacts. Second, the amount of surface data available will most likely significantly increase in the forthcoming years and appropriate methodologies to infer ice thickness distribution will have to be developed. The paper is well structured and written; I therefore recommend its publication in the best delay.*

**AR:** We thank the reviewer for this very positive appreciation and are particularly pleased by his/her judgement regarding the relevance of this community effort.

――――――

**RC:** *I must confess that I had one slight frustration reading the manuscript: there is not that many recommendations on further direction of developments. I clearly have in mind how difficult this could be in such a large community effort with a lot of variety in the approaches used and the incomplete realization of the whole set of experiments by most of the models participating. I have one suggestion to circumvent this difficulty, which I believe would also improve the readability of the paper and facilitate its impact on non-specialists. 17 different models have been used, and according to the authors, they could be classified in 4 categories. In the manuscript the description and discussion are only viewed from a single model point of view or from the entire set of models. I think that approaching also the manuscript using these 4 categories would strengthen the paper. Some suggestions regarding that point:*

**AR:** This comment is very much in line with the major issue raised by Reviewer #1. As explained above, we addressed this point by classifying the participating approaches into five categories. This decision led us to re-organize Section 4 completely, and the categories are now used to guide the visual presentation of the results and part of the discussion.

In addition, we slightly re-arranged our conclusion section. More importantly, we explicitly summarized our recommendations at the end of the section by introducing the following statement:

**Section 6, last Paragraph**
"To summarize: In order to improve available thickness estimates for glacier and ice caps, we make the following recommendations:

- Ensemble-methods comprising a variety of independent, physically-based approaches should be considered. This is likely to be a more effective strategy than focusing on one individual approach.
- Models should be extended to take observational uncertainty into account. The Bayesian framework used by Brinkerhoff et al. (2016), for example, showed promising results in this respect.
- The increasing availability of surface ice-flow velocity data (e.g. **?**) should be exploited. In this context, the previously-mentioned necessity of accounting for observational uncertainty is crucial.
- The way individual models treat ice divides has to be improved. This is important when addressing ice caps and glacier complexes, and to ensure consistency between subsurface topographies of adjacent ice masses.
- Efforts for centralizing available ice thickness measurements should be strengthened. Initiatives such as the GlaThiDa database launched by the World Glacier Monitoring Service are essential for new generations of ice thickness models to be developed and validated."

**RC:** *Section 4, is rather technical and probably gives not enough details for specialists and too many for non-specialists. I would recommend arranging section 4 in order to present the philosophy of each type/category of method and to add in the supplementary materials required details for each model. Regarding that latter aspect, I would strongly encouraged unpublished approaches to give more details as information are today to my opinion very limited.*

**AR:** See previous answer and also answers to Reviewer #1: We redesigned Section 4, which – as suggested by the reviewer – is now centred around a description of the principles underlying the five categories. For what the description of the unpublished models is concerned, we moved the description to the Supplementary material (see Supplementary Section S1). This allowed us to present the unpublished methods in much greater detail and we are confident that this is now sufficient for properly understanding the methods.

**RC:** *In more or less all figures and tables (and particularly in table 1, table 2, table 4, Figure 3, Figure 7) the 17 models are presented in alphabetic order. I would recommend grouping them along the 4 categories. Color code of figures 4, 5 might also be rearranged to try to give the opportunity to the reader of disentangling the approach used behind the models.*

**AR:** Following the reviewer's suggestion, we changed the sequence (Table 2) and colors (Figures 2, 3, 4, 5, and 7) by which the individual modes are presented to reflect the five categories introduced in Section 4. The categories are also referred to when presenting the individual rankings (Table 3 and 4). We are confident that this achieved the anticipated effect.

**RC:** *Are they pattern emerging regarding these categories? Systematic bias for a given category for some specific configuration (glaciers, ice caps, synthetic)? In other words is there a family of approach that is more suited for a given configuration or not? I think this should be discussed. My guess is that if there were some obvious ones it would have been already discussed. However, mentioning that today we are not able to discriminate whether one family of approach seems more appropriate than another is also a result that I think deserve to be mentioned.*

**AR:** No, the reviewer is correct in stating that no clear pattern emerges. To better clarify this, we introduced a new paragraph at the beginning of Section 5 (cf. reply to "comment 3" by Reviewer #1), and propose an additional figure in the supplementary material in which the deviations of all models of a given category are pooled (Supplementary Figure S3). Similarly, we clarify the same aspect when introducing the results of the two ranking schemes that we propose. We do this with the following wording:

**Section 5.3, Paragraph 5**

"Similarly as noted during the discussion of the last section (Sec. 5), the rankings do not suggest a performance advantage in any of the five model categories introduced in Section 4."

**RC:** *I would further encourage having a perennial repository of all the datasets required for people to confront future approaches with the results already computed (i.e. set up and results). This is mentioned as an intention in the conclusion, but a clear link to an already established database would be much better. Ideally, this database should be reviewed with the manuscript.*

**AR:** Providing such a perennial repository is indeed our intention. A link to a repository

was missing in the first version of our manuscript as we anticipate to assign a DOI to it. For the moment, we provide a temporary link (see below) that will expire once the publication is accepted. In the final version of the paper, a permanent DOI will be disclosed.

**Section 6, last sentence**

"All data used within ITMIX, as well as the individual solutions submitted by the participating models, can be found at   [http://3cm.ch/ITMIX_full_dataset.zip](http://3cm.ch/ITMIX_full_dataset.zip)"

**References**

Bahr, D., Meier, M., and Peckham, S. (1997). The physical basis of glacier volume-area scaling. *Journal of Geophysical Research*, 102(B9):20355–20362. doi: 10.1029/97JB01696.

Bahr, D., Pfeffer, W., and Kaser, G. (2015). A review of volume-area scaling of glaciers. *Reviews of Geophysics*, 53(1):95–140. doi: 10.1002/2014RG000470.

Brinkerhoff, D., Aschwanden, A., and Truffer, M. (2016). Bayesian inference of sub-glacial topography using mass conservation. *Frontiers in Earth Science*, 4(8):1–15. doi: 10.3389/feart.2016.00008.

Gantayat, P., Kulkarni, A., and Srinivasan, J. (2014). Estimation of ice thickness using surface velocities and slope: case study at Gangotri Glacier, India. *Journal of Glaciology*, 60(220):277–282. doi: 10.3189/2014JoG13J078.

Kamb, B. and Echelmeyer, K. A. (1986). Stress-gradient coupling in glacier flow: I. longitudinal averaging of the influence of ice thickness and surface slope. *Journal of Glaciology*, 32(111):267–284.

Raftery, A., Balabdaoui, F., Gneiting, T., and Polakowski, M. (2005). Using bayesian model averaging to calibrate forecast ensembles. *Monthly Weather Review*, 133:1155–1174. doi: 10.1175/MWR2906.1.

---

## Author Response (AR2)

**TC-2016-250**
**Production-files upload**

Daniel Farinotti, and the ITMIX consortium

To whom it may concern,

We would like to thank the editor for the positive decision and for spotting the minor issues requiring technical correction. All corrections have been performed and the files for production prepared according to the journal instructions.

Note that the link under which our data are stored (mentioned at the very end of the manuscript) is a temporary link. Since we want our data to make reference to the article itself, we would need a DOI for it. Once the article-DOI is defined, we will add it to our dataset, and provide a DOI for the latter.

In behalf of all authors
Daniel Farinotti